# Can Adaptive Gradient Methods Converge under Heavy-Tailed Noise?
# A Case Study of AdaGrad

**Zijian Liu** [1]

## Abstract

Many tasks in modern machine learning are observed to involve heavy-tailed gradient noise during the optimization process. To manage this realistic and challenging setting, new mechanisms, such as gradient clipping and gradient normalization, have been introduced to ensure the convergence of first-order algorithms. However, adaptive gradient methods, a famous class of modern optimizers that includes popular `Adam` and `AdamW`, often perform well even without any extra operations mentioned above. It is therefore natural to ask whether adaptive gradient methods can converge under heavy-tailed noise without any algorithmic changes. In this work, we take the first step toward answering this question by investigating a special case, `AdaGrad`, the origin of adaptive gradient methods. We provide the first provable convergence rate for `AdaGrad` in non-convex optimization when the tail index $p$ satisfies $4/3 < p \leq 2$. Notably, this result is achieved without requiring any prior knowledge of $p$ and is hence adaptive to the tail index. In addition, we develop an algorithm-dependent lower bound, suggesting that the existing minimax rate for heavy-tailed optimization is not attainable by `AdaGrad`. Lastly, we consider `AdaGrad-Norm`, a popular variant of `AdaGrad` in theoretical studies, and show an improved rate that holds for any $1 < p \leq 2$ under an extra mild assumption.

## 1. Introduction

The heavy-tailed phenomenon has been widely observed in the optimization process for modern machine learning tasks across various domains (Simsekli et al., 2019; Garg et al., 2021; Battash et al., 2024), and is particularly prevalent when training attention-based models (Vaswani et al., 2017; Zhang et al., 2020b; Ahn et al., 2024). More concretely, it refers to the gradient noise (i.e., the difference between the stochastic gradient and the true gradient) having only a finite $p$-th moment for some $p \in (1, 2]$, rather than satisfying the classical finite-variance condition (i.e., $p = 2$) commonly adopted in the stochastic optimization literature (Bottou et al., 2018; Lan, 2020).

For first-order methods, two approaches are known to guarantee provable convergence in non-convex optimization under heavy-tailed noise. One way is based on gradient clipping (i.e., Clipped Stochastic Gradient Descent (`Clipped SGD`)), which artificially limits the norm of the stochastic gradient within a user-specific threshold (Zhang et al., 2020b; Liu et al., 2023c; Sadiev et al., 2023; Nguyen et al., 2023). The other kind relies on gradient normalization, as recently discovered by Hübler et al. (2025); Liu & Zhou (2025); Sun et al. (2025), who show that the normalization mechanism in Normalized Stochastic Gradient Descent (with Momentum) (`NSGD(M)`) (Nesterov, 1984; Cutkosky & Mehta, 2020) can also successfully tackle heavy-tailed noise. Interestingly, Hübler et al. (2025); Liu & Zhou (2025) also provided a convergence rate achieved without any prior knowledge of problem-dependent parameters, which is the first in the literature. In particular, the feature of not requiring any information on the tail index $p$ highlights a key advantage of using `NSGD(M)`.

Despite the progress, an important gap still remains between theory and practice. Specifically, the above results cannot cover a well-known class of algorithms widely used in practice, namely, adaptive gradient methods, whose empirical effectiveness has been repeatedly demonstrated in the training of neural networks, including large language models. This class includes `AdaGrad`, introduced in two pioneering works (McMahan & Streeter, 2010; Duchi et al., 2011), followed by `RMSProp` (Tieleman et al., 2012), then more practical algorithms nowadays, such as `Adam` (Kingma & Ba, 2014) and `AdamW` (Loshchilov & Hutter, 2019), along with many further variants. In other words, the reason for the strong performance of adaptive gradient methods under heavy-tailed noise remains largely unclear and warrants further exploration.

---

[1]Stern School of Business, New York University. Correspondence to: Zijian Liu <zl3067@stern.nyu.edu>.

*Proceedings of the $43^{rd}$ International Conference on Machine Learning*, Seoul, South Korea. PMLR 306, 2026. Copyright 2026 by the author(s).

To the best of our knowledge, only one recent work by Chezhegov et al. (2025) studies AdaGrad- and Adam-based methods under heavy-tailed noise. However, their work has some limitations that we will discuss below.

First and foremost, the main results presented in Chezhegov et al. (2025) are established for *delayed* adaptive gradient algorithms introduced by Li & Orabona (2019), in which the stepsize constructed at the $t$-th iteration depends only on stochastic gradients up to time $t - 1$, rather than time $t$. It is known that the delayed variant requires a different style of theoretical analysis, since it makes the stepsize and the stochastic gradient conditionally independent. Moreover, this modification is rarely implemented in practice. Consequently, the applicability of their theoretical guarantees to practical algorithms is limited, leaving the gap open.

Next, Theorem 3.3 in Chezhegov et al. (2025) is the only result in their work not for the delayed setting. However, it still cannot directly apply to AdaGrad and Adam, as the algorithms considered there do not employ the coordinate-wise update and additionally require gradient clipping. Moreover, from a theoretical perspective, their Theorem 3.3 has two shortcomings. One is the extra assumption of boundedness for objective functions, which is stronger than the standard lower boundedness condition in the non-convex optimization literature, thereby reducing the generality. The other is requiring the value of problem-dependent parameters (e.g., the tail index $p$) as inputs to ensure convergence, contradicting the original purpose of adaptive gradient methods.

Therefore, Chezhegov et al. (2025) cannot fully explain the empirical success of adaptive gradient methods under heavy-tailed noise, leaving a large room for further improvement.

Motivated by the above discussion, we are naturally led to the following question:

*Can adaptive gradient methods converge under heavy-tailed noise in non-convex optimization, without any algorithmic modifications, nonstandard assumptions, or prior knowledge of problem-dependent parameters?*

### 1.1. Our Contributions

In this work, we take the first step toward answering the above question through a case study of AdaGrad, the origin of adaptive gradient methods, and make the following contributions:

- In Theorem 3.1, we show that AdaGrad provably converges at a rate of $\tilde{\mathcal{O}}(1/T^{\frac{3p-4}{4p}})$ after $T$ iterations under heavy-tailed noise in non-convex optimization, which is meaningful when $p \in (4/3, 2]$. To the best of our knowledge, this result provides the first theoretical justification for the convergence of AdaGrad in the heavy-tailed setting, without any algorithmic modifi-

cations, nonstandard assumptions, or prior knowledge of problem-dependent parameters, thereby (partially) confirming the question asked earlier.

- In Theorem 3.3, we establish the first algorithm-dependent lower bound for AdaGrad in heavy-tailed non-convex optimization, explicitly capturing the dependence on the input learning rate. Our result suggests that the existing minimax rate for heavy-tailed non-convex optimization (Zhang et al., 2020b; Liu & Zhou, 2025; Liu, 2026) is generally not attainable by AdaGrad. Even in the special case of $p = 2$, our bound also improves upon the existing lower bound for AdaGrad proved by Jiang et al. (2025).

Moreover, we study AdaGrad-Norm (Ward et al., 2019), a popular variant of AdaGrad in theoretical research.

- In Theorem 4.2, we prove that AdaGrad-Norm converges at a rate of $\mathcal{O}(1/T^{\frac{p-1}{2p}})$ under the additional assumption of bounded objectives, as considered in Chezhegov et al. (2025). This rate never becomes vacuous for any $p \in (1, 2]$ and also does not require any prior information on problem-dependent parameters.

- In Theorem C.1, we further provide an upper bound of $\tilde{\mathcal{O}}(1/T^{\frac{3p-4}{4p}})$ for AdaGrad-Norm without the extra boundedness assumption, matching the rate for AdaGrad in terms of $T$ in the same setting.

Finally, in Section 5, we also discuss the limitations of our work and outline possible future directions.

### 1.2. Related Work

We first review the literature on adaptive gradient methods.

**Adaptive gradient methods.** The study of adaptive gradient methods traces back to two pioneering works (McMahan & Streeter, 2010; Duchi et al., 2011), independently introducing the first adaptive gradient method AdaGrad. Later on, RMSProp, a combination of AdaGrad and the mean square estimation technique, was proposed by Tieleman et al. (2012). By further incorporating momentum into RMSProp, Adam was developed in the seminal work of Kingma & Ba (2014). Furthermore, Loshchilov & Hutter (2019) introduced decoupled weight decay into Adam, resulting in a new algorithm now known as AdamW. In addition to these methods, numerous variants exist in the literature, for example, AMSGrad (Reddi et al., 2018) and Adafactor (Shazeer & Stern, 2018).

Although many adaptive gradient methods were originally designed to guarantee sublinear regret in online convex optimization (e.g., AdaGrad and Adam), they have been

observed to perform well across a wide range of modern machine learning tasks, which are however typically non-convex. As far as we know, Ward et al. (2019) established the first provable rate for `AdaGrad-Norm` (a simple variant of `AdaGrad`), serving as a cornerstone of theoretical studies for adaptive gradient methods in non-convex optimization. Subsequently, a large body of work has developed comprehensive studies of the convergence theory for different adaptive gradient methods, including `AdaGrad`, `RMSProp`, `Adam`, and `AdamW` (or their variants), in both deterministic and stochastic non-convex optimization (Zaheer et al., 2018; De et al., 2018; Zou et al., 2019; Shi et al., 2021; Défossez et al., 2022; Kavis et al., 2022; Faw et al., 2022; Zhang et al., 2022; Faw et al., 2023; Liu et al., 2023b; Attia & Koren, 2023; YANG et al., 2023; Wang et al., 2023a;b; 2024; Hong & Lin, 2024; Liu et al., 2025; Jiang et al., 2025; Zhang et al., 2025; Li et al., 2025). However, under the classical finite-variance assumption (or similar conditions), the existing theory does not reflect a substantial advantage of adaptive gradient methods over `SGD`, as they share the same convergence rate $\tilde{\mathcal{O}}(1/T^{\frac{1}{4}})$ in terms of the time horizon $T$ to find a stationary point.

As for lower bounds of adaptive gradient methods, the only result we are aware of is for `AdaGrad`[1] under $p = 2$ given by Jiang et al. (2025), which establishes a complexity lower bound of $\Omega((\Delta L \epsilon^{-2} + \Delta L \sigma^2 \epsilon^{-4}) \ln(\Delta L \epsilon^{-2}))$[2] to find an $\epsilon$-stationary point, where $\Delta$ denotes the initial function value gap, $L > 0$ is the smoothness parameter, and $\sigma \geq 0$ characterizes the noise level. This bound is larger than the minimax rate for stochastic non-convex optimization under the finite-variance condition (Arjevani et al., 2023). However, it is not algorithm-dependent, since it fails to capture the dependence on the input learning rate.

Next, we provide a basic background on smooth non-convex optimization under heavy-tailed noise.

**Upper bound under heavy-tailed noise.** For clipping-based algorithms (e.g., `Clipped SGD`), several works have established the optimal convergence rate $\mathcal{O}(1/T^{\frac{p-1}{3p-2}})$ (or $\tilde{\mathcal{O}}(1/T^{\frac{p-1}{3p-2}})$) in both expectation and high probability (Zhang et al., 2020b; Cutkosky & Mehta, 2021; Liu et al., 2023c; Nguyen et al., 2023). Such rates are always derived based on the prior value of problem-dependent parameters, in particular, the tail index $p$. Recent works (Hübler et al., 2025; Liu & Zhou, 2025; Sun et al., 2025) further show that the normalization-based method (i.e., `NSGD(M)`) can also

achieve the optimal rate of $\mathcal{O}(1/T^{\frac{p-1}{3p-2}})$ when prior information about the problem is available. Moreover, Hübler et al. (2025); Liu & Zhou (2025) also prove that `NSGD(M)` can converge at a rate of $\mathcal{O}(1/T^{\frac{p-1}{2p}})$ without any prior information.

**Lower bound under heavy-tailed noise.** For non-convex optimization under heavy-tailed noise, to find an $\epsilon$-stationary point in expectation, any (possibly randomized) algorithm must query at least $\Omega(\Delta L \epsilon^{-2} + \Delta L \sigma^{\frac{p}{p-1}} \epsilon^{-\frac{3p-2}{p-1}})$ stochastic gradients (Zhang et al., 2020b; Liu & Zhou, 2025; Liu, 2026), where $p \in (1, 2]$ is the tail index, $\Delta$ denotes the initial function value gap, $L > 0$ is the smoothness parameter, and $\sigma \geq 0$ characterizes the noise level. However, the algorithm (e.g., `Clipped SGD`) that can achieve this minimax lower bound often requires prior knowledge of all these parameters, which is usually not practical.

Recently, Hübler et al. (2025) establishes the first algorithm-dependent lower bound for `NSGD` in one-dimensional optimization that captures the dependence on the stepsize and batch size when problem-dependent parameters are unknown in advance. In particular, their result can be simplified to an $\Omega((\Delta^4 + L^4)\epsilon^{-4} + \sigma^{\frac{2p}{p-1}} \epsilon^{-\frac{2p}{p-1}})$ lower bound for `NSGD` when no prior information is available.

## 2. Preliminary

**Notation.** In this paper, scalars and vectors are denoted by regular and bold fonts, respectively. $\mathbb{N}$ is the set of natural numbers (excluding 0). $[n] \triangleq \{1, \ldots, n\}, \forall n \in \mathbb{N}$. $\lceil \cdot \rceil$ is the ceiling function. For $a > 1$, we denote by $\bar{a}$ the conjugate of $a$ (i.e., $\frac{1}{a} + \frac{1}{\bar{a}} = 1$). $\mathbb{R}^d_{>0}$ (resp. $\mathbb{R}^d_{\geq 0}$) is the set of vectors in $\mathbb{R}^d$ whose coordinates are all positive (resp. non-negative). Given $\boldsymbol{\Lambda} \in \mathbb{R}^d_{>0}$, its induced inner product and norm are $\langle \boldsymbol{x}, \boldsymbol{y} \rangle_{\boldsymbol{\Lambda}} \triangleq \sum_{i=1}^d \boldsymbol{x}_i \boldsymbol{\Lambda}_i \boldsymbol{y}_i$ and $\|\boldsymbol{x}\|_{\boldsymbol{\Lambda}} \triangleq \sqrt{\langle \boldsymbol{x}, \boldsymbol{x} \rangle_{\boldsymbol{\Lambda}}}$, respectively. In addition, we use the following shorthands $(\boldsymbol{xy})_i \triangleq \boldsymbol{x}_i \boldsymbol{y}_i, \boldsymbol{x}^2 \triangleq \boldsymbol{xx}, (\frac{\boldsymbol{x}}{\boldsymbol{y}})_i \triangleq \frac{\boldsymbol{x}_i}{\boldsymbol{y}_i}$, and $(\sqrt{\boldsymbol{z}})_i = \sqrt{\boldsymbol{z}_i}$ for any $i \in [d]$ and $\boldsymbol{x}, \boldsymbol{y} \in \mathbb{R}^d, \boldsymbol{z} \in \mathbb{R}^d_{\geq 0}$. Given a differentiable function $h$, $\nabla_i h$ is the partial derivative w.r.t. the $i$-th coordinate.

**Objective.** This work studies the optimization problem

$$\min_{\boldsymbol{x} \in \mathbb{R}^d} f(\boldsymbol{x}),$$

where $f : \mathbb{R}^d \to \mathbb{R}$ is differentiable and possibly non-convex. Since finding a global optimal solution can be computationally intractable, we shift the focus to minimizing $\|\nabla f(\boldsymbol{x})\|$ as in the non-convex optimization literature.

**Assumptions.** We first make the following assumptions.
**Assumption 2.1** (Lower boundedness)**.** *The objective satisfies* $f_\star \triangleq \inf_{\boldsymbol{x} \in \mathbb{R}^d} f(\boldsymbol{x}) > -\infty$.

---

[1] For `AdaGrad`, Crawshaw & Liu (2025) also proved a lower bound result that is not directly comparable to Jiang et al. (2025) due to a different setting.

[2] For convenience, we only state their result in the one-dimensional case. Jiang et al. (2025) also established a lower bound in the high-dimensional case.

---

**Algorithm 1** AdaGrad (McMahan & Streeter, 2010; Duchi et al., 2011)

---

**Input:** initial point $\boldsymbol{x}_1 \in \mathbb{R}^d$, learning rate $\gamma > 0$, hyperparameter $\lambda > 0$
**Initialization:** $\boldsymbol{v}_0 = \boldsymbol{0}$
**for** $t = 1$ **to** $T$ **do**
    $\boldsymbol{v}_t = \boldsymbol{v}_{t-1} + \boldsymbol{g}_t^2$
    $\boldsymbol{x}_{t+1} = \boldsymbol{x}_t - \frac{\gamma}{\lambda + \sqrt{\boldsymbol{v}_t}} \boldsymbol{g}_t$
**end for**

---

**Assumption 2.2** (Smoothness). $\exists \boldsymbol{L} \in \mathbb{R}_{>0}^d$ such that $|f(\boldsymbol{x}) - f(\boldsymbol{y}) - \langle \nabla f(\boldsymbol{y}), \boldsymbol{x} - \boldsymbol{y} \rangle| \leq \frac{1}{2} \|\boldsymbol{x} - \boldsymbol{y}\|_{\boldsymbol{L}}^2$ for any $\boldsymbol{x}, \boldsymbol{y} \in \mathbb{R}^d$, or equivalently, $\|\nabla f(\boldsymbol{x}) - \nabla f(\boldsymbol{y})\|_{1/\boldsymbol{L}} \leq \|\boldsymbol{x} - \boldsymbol{y}\|_{\boldsymbol{L}}$ for any $\boldsymbol{x}, \boldsymbol{y} \in \mathbb{R}^d$.

Assumption 2.1 is standard in the literature. Assumption 2.2 is the coordinate-wise counterpart of the classical smoothness condition. This kind of fine-grained smoothness has been studied before, for example, in Bernstein et al. (2018; 2019); Liu et al. (2023a); Jiang et al. (2025); Liu et al. (2025).

Since we consider stochastic optimization, given a point $\boldsymbol{x}_t \in \mathbb{R}^d$ at the $t$-th iteration, $\boldsymbol{g}_t$ hereinafter denotes the stochastic gradient queried at $\boldsymbol{x}_t$. $\mathcal{F}_t \triangleq \sigma(\boldsymbol{g}_1, \ldots, \boldsymbol{g}_t)$ is the natural filtration, and $\mathbb{E}_t[\cdot] \triangleq \mathbb{E}[\cdot \mid \mathcal{F}_t]$ represents the conditional expectation given $\mathcal{F}_t$.

Our analysis also relies on the next two assumptions.

**Assumption 2.3** (Unbiased gradient). *The stochastic gradient satisfies* $\mathbb{E}_{t-1}[\boldsymbol{g}_t] = \nabla f(\boldsymbol{x}_t)$.

**Assumption 2.4** (Heavy-tailed noise). $\exists p \in (1, 2]$ *and* $\boldsymbol{\sigma} \in \mathbb{R}_{\geq 0}^d$ *such that* $\mathbb{E}_{t-1}\left[|\boldsymbol{\xi}_{t,i}|^p\right] \leq \boldsymbol{\sigma}_i^p, \forall i \in [d]$, *where* $\boldsymbol{\xi}_{t,i} \triangleq \boldsymbol{g}_{t,i} - \nabla_i f(\boldsymbol{x}_t)$.

*Remark* 2.5. Our proof strategy still works when replacing Assumption 2.4 with a weaker version: $\mathbb{E}_{t-1}\left[|\boldsymbol{\xi}_{t,i}|^{p_i}\right] \leq \boldsymbol{\sigma}_i^{p_i}, \forall i \in [d]$, where $p_i \in (1, 2]$ can take different values for different coordinates. However, to make the work more concise, we keep the current simpler version.

Assumption 2.3 is a common condition in stochastic optimization. Assumption 2.4 appears in Chezhegov et al. (2025) and differs slightly from the popular one for heavy-tailed noise in the literature, which typically takes the form of $\mathbb{E}_{t-1}\left[\|\boldsymbol{\xi}\|_2^p\right] \leq \sigma^p$ for some $\sigma \geq 0$. It can be interpreted as a coordinate-wise version of heavy-tailed noise, which is natural in our setting since AdaGrad employs a coordinate-specific update rule. Moreover, Assumption 2.4 also generalizes the coordinate-wise finite-variance assumption considered in prior works (Bernstein et al., 2018; 2019; Jiang et al., 2025; Liu et al., 2025; Li et al., 2025).

*Remark* 2.6. A seemingly similar condition to Assumption 2.4, but in fact fundamentally different, is Assumption 2 in Zhang et al. (2020b), where each coordinate of the stochastic gradient $\boldsymbol{g}_{t,i}$ is assumed to have a finite $p$-th moment, rather than each coordinate of the noise $\boldsymbol{\xi}_{t,i}$.

## 3. AdaGrad under Heavy-Tailed Noise

The optimizer focused on in this work, AdaGrad, is described in Algorithm 1. AdaGrad was independently introduced in two pioneering works by McMahan & Streeter (2010) and Duchi et al. (2011). The key mechanism of AdaGrad is to dynamically adjust the stepsize, i.e., $\frac{\gamma}{\lambda + \sqrt{\boldsymbol{v}_t}}$, based on all stochastic gradient information up to the current iteration in a coordinate-wise manner.

### 3.1. Upper Bound

We present the first provable upper bound for AdaGrad under heavy-tailed noise in the following Theorem 3.1, the proof of which is deferred to Appendix A.

**Theorem 3.1.** *Under Assumptions 2.1, 2.2, 2.3, and 2.4, let* $\Delta \triangleq f(\boldsymbol{x}_1) - f_\star$, *then for any* $\gamma > 0$ *and* $\lambda > 0$, AdaGrad *(Algorithm 1) guarantees*

$$\mathbb{E}\left[\frac{1}{T}\sum_{t=1}^{T}\|\nabla f(\boldsymbol{x}_t)\|_1\right]$$
$$\leq \mathcal{O}\left(\frac{A}{\sqrt{T}} + \frac{C\|\boldsymbol{\sigma}\|_1}{T^{\frac{p-1}{p}}} + \frac{\sqrt{B\|\boldsymbol{\sigma}\|_1}}{T^{\frac{p-1}{2p}}} + \frac{\sqrt{C}\|\boldsymbol{\sigma}\|_1}{T^{\frac{3p-4}{4p}}}\right),$$

*where* $A \triangleq d\lambda + \frac{\Delta}{\gamma} + \gamma\|\boldsymbol{L}\|_1 \ln K_T$, $B \triangleq \frac{\Delta}{\gamma} + \gamma\|\boldsymbol{L}\|_1 \ln K_T$, $C \triangleq \ln^{\frac{1}{p}+\frac{1}{2}} K_T$, *and* $K_T$ *is in the order of* $\mathrm{poly}(T, \|\boldsymbol{L}\|_1, \|\boldsymbol{L}\|_\infty, \|\boldsymbol{\sigma}\|_\infty, \|\nabla f(\boldsymbol{x}_1)\|_\infty, \gamma, 1/\lambda)$.

*Remark* 3.2. For the precise definition of $K_T$, we refer the reader to Theorem A.1.

To the best of our knowledge, Theorem 3.1 provides the first convergence rate for AdaGrad under heavy-tailed noise when the tail index $p$ lies in the regime $(4/3, 2]$. Remarkably, this result holds under the standard assumptions without requiring any algorithmic modifications or prior knowledge of problem-dependent parameters. In contrast, the minimax rate $\Theta(1/T^{\frac{p-1}{3p-2}})$ in the literature has been achieved only when problem-dependent parameters, particularly the value of $p$, are assumed to be known.

The important feature of Theorem 3.1 is its adaptivity. First, it is adaptive to the tail index $p$. In other words, AdaGrad can automatically adapt to the largest admissible value of $p$ without any tuning. In particular, in the boundary case $p = 2$, Theorem 3.1 recovers the well-known $\tilde{\mathcal{O}}(1/T^{\frac{1}{4}})$

rate of `AdaGrad`, matching existing results in the literature (Wang et al., 2023b; Jiang et al., 2025). Next, the convergence rate is also adaptive to the noise level $\|\boldsymbol{\sigma}\|_1$, as in the existing analysis of `AdaGrad` under $p = 2$. More precisely, Theorem 3.1 recovers the best possible rate $\tilde{\mathcal{O}}(1/T^{\frac{1}{2}})$ in the noiseless case when $\boldsymbol{\sigma} = \boldsymbol{0}$ (or $\|\boldsymbol{\sigma}\|_1$ is sufficiently small to be negligible). In summary, Theorem 3.1 is the first time demonstrating that `AdaGrad` is simultaneously adaptive to both the tail index $p$ and the noise level $\|\boldsymbol{\sigma}\|_1$.

Therefore, we would like to highlight that, as far as we know, this is the first concrete theoretical evidence supporting the advantage of adaptive gradient methods over `SGD` under heavy-tailed noise.

**Novel technique in the analysis.** Now, we discuss the novel part of our proof. Following the literature on `AdaGrad`, our analysis also employs a proxy stepsize proposed by Ward et al. (2019) (also known as decorrelated stepsize (Faw et al., 2022)), i.e., a vector $\boldsymbol{w}_t \in \mathbb{R}_{\geq 0}^d$ that is predictable (in other words, $\boldsymbol{w}_t \in \mathcal{F}_{t-1}$) to approximate $\boldsymbol{v}_t$. As far as we know, there are typically two choices for $\boldsymbol{w}_t$ in the literature. One is to set $\boldsymbol{w}_t \triangleq \boldsymbol{v}_{t-1} + (\nabla f(\boldsymbol{x}_t))^2 + \boldsymbol{\sigma}^2$ (Ward et al., 2019). The other is to set $\boldsymbol{w}_t \triangleq \boldsymbol{v}_{t-1}$ (Wang et al., 2023b).

The technical contribution of our analysis is to further generalize the first kind of proxy stepsize. Concretely, we set

$$\boldsymbol{w}_t \triangleq \boldsymbol{v}_{t-1} + (\nabla f(\boldsymbol{x}_t))^2 + \boldsymbol{c}^2,$$

where $\boldsymbol{c} \in \mathbb{R}_{\geq 0}^d$ is a free parameter that can be determined at the end of the proof. Though this extension seems simple, it is in fact critical to the analysis. If one simply picks $\boldsymbol{c} = \boldsymbol{\sigma}$ as in many existing works (e.g., Jiang et al. (2025)), the best possible rate we can derive is in the order of $\tilde{\mathcal{O}}(1/T^{\frac{2p-3}{2p}})$, which is always worse than the rate $\tilde{\mathcal{O}}(1/T^{\frac{3p-4}{4p}})$ given in Theorem 3.1. Instead, in our proof, we pick

$$\boldsymbol{c}_i \triangleq \frac{\boldsymbol{\sigma}_i T^{\frac{1}{2}-\frac{1}{p}}}{D_{T,i}^{\frac{1}{2}-\frac{1}{p}}}, \forall i \in [d],$$

where

$$D_{T,i} \triangleq 2\ln\left(1 + \frac{\boldsymbol{\sigma}_i T^{\frac{1}{p}} + \mathbb{E}\left[\sqrt{\sum_{t=1}^{T}(\nabla_i f(\boldsymbol{x}_t))^2}\right]}{\lambda/\sqrt{2}}\right).$$

This choice turns out to be the key to obtaining the rate $\tilde{\mathcal{O}}(1/T^{\frac{3p-4}{4p}})$ stated in Theorem 3.1.

In particular, our choice of $\boldsymbol{c}$ degenerates to $\boldsymbol{\sigma}$ when $p = 2$, since $\frac{1}{2} - \frac{1}{p} = 0$ due to $\bar{p} = \frac{p}{p-1} = 2$, meaning that our $\boldsymbol{w}_t$ naturally recovers the popular proxy stepsize $\boldsymbol{w}_t = \boldsymbol{v}_{t-1} + (\nabla f(\boldsymbol{x}_t))^2 + \boldsymbol{\sigma}^2$ in the classical finite-variance situation. Thus, our technique is indeed a novel generalization.

For more details of this technique, see Appendix A.

### 3.2. Algorithm-Dependent Lower Bound

In this subsection, we provide the first algorithm-dependent lower bound for `AdaGrad` under heavy-tailed non-convex optimization.

**Theorem 3.3.** *Let $d = 1$, for any given $\Delta > 0$, $L > 0$, $p \in (1, 2]$, $\sigma \geq 0$, $0 < \epsilon \leq \sqrt{2\Delta L}$, $x_1 \in \mathbb{R}$, $\gamma > 0$, there exists a function $f : \mathbb{R} \to \mathbb{R}$ associated with a stochastic gradient oracle $g$ satisfying Assumptions 2.1 (and also $f(x_1) - \inf_{x \in \mathbb{R}} f(x) \leq \Delta$), 2.2 (with parameter $L$), 2.3, and 2.4 (with parameters $p$ and $\sigma$). Moreover, if using* `AdaGrad` *(Algorithm 1), with initial point $x_1$, learning rate $\gamma$, and hyperparameter $\lambda = 0$, to optimize $f$ by interacting with $g$, one must use at least*

$$\Omega\left(\frac{\frac{\Delta^2}{\gamma^2} + \gamma^2 L^2 \ln^2(\frac{\gamma L}{\epsilon})}{\epsilon^2} + \frac{\left(\frac{\Delta^2}{\gamma^2} + \gamma^2 L^2 \ln^2(\frac{\gamma L}{\epsilon})\right)\sigma^{\frac{p}{p-1}}}{\epsilon^{\frac{3p-2}{p-1}}}\right)$$

*iterations to make $\mathbb{E}\left[\frac{1}{T}\sum_{t=1}^{T}|f'(x_t)|\right] \leq \mathcal{O}(\epsilon)$ for small enough $\epsilon$.*

*Remark* 3.4. The requirement of $\lambda = 0$ is only for simplicity. See Theorem B.1 for the full version that allows $\lambda \geq 0$.

*Remark* 3.5. For simplicity, we restrict our attention to the case $d = 1$. Following the proof of Theorem 4 in Jiang et al. (2025), Theorem 3.3 can be extended to the high-dimensional setting.

Theorem 3.3 provides the first algorithm-dependent lower bound for `AdaGrad` in the case $d = 1$, explicitly capturing the dependence on the input learning rate.

To better understand Theorem 3.3, let us first consider $p = 2$, corresponding to the finite-variance setting. In this case, Theorem 3.3 degenerates to

$$\Omega\left(\frac{\frac{\Delta^2}{\gamma^2} + \gamma^2 L^2 \ln^2(\frac{\gamma L}{\epsilon})}{\epsilon^2} + \frac{\left(\frac{\Delta^2}{\gamma^2} + \gamma^2 L^2 \ln^2(\frac{\gamma L}{\epsilon})\right)\sigma^2}{\epsilon^4}\right).$$

We claim that the above bound improves upon the existing one-dimensional lower bound

$$\Omega\left(\frac{\Delta L \ln(\frac{\Delta L}{\epsilon^2})}{\epsilon^2} + \frac{\Delta L \sigma^2 \ln(\frac{\Delta L}{\epsilon^2})}{\epsilon^4}\right)$$

established by Jiang et al. (2025) (see their Lemma 16). Indeed, note that

$$\inf_{\gamma > 0} \frac{\Delta^2}{\gamma^2} + \gamma^2 L^2 \ln^2\left(\frac{\gamma L}{\epsilon}\right)$$

$$= \inf_{\eta > 0} \frac{\Delta L}{2}\left(\frac{1}{\eta} + \eta \ln^2\left(\frac{2\eta\Delta L}{\epsilon^2}\right)\right) \geq \frac{\Delta L}{2}\ln\left(\frac{2\Delta L}{\epsilon^2}\right),$$

where we substitute $\gamma = \sqrt{2\eta\Delta/L}$ in the first step and apply Lemma B.5 in the second step. Therefore, our lower

---

**Algorithm 2** AdaGrad-Norm (Ward et al., 2019)

---

**Input:** initial point $x_1 \in \mathbb{R}^d$, learning rate $\gamma > 0$, hyperparameter $\lambda > 0$
**Initialization:** $v_0 = 0$
**for** $t = 1$ **to** $T$ **do**
    $v_t = v_{t-1} + \|g_t\|_2^2$
    $x_{t+1} = x_t - \frac{\gamma}{\lambda + \sqrt{v_t}} g_t$
**end for**

---

bound for AdaGrad is strictly more refined than the best-known one in the literature.

For general $p \in (1, 2]$, Theorem 3.3 is saying that, without any prior information on $\Delta$ and $L$, AdaGrad is impossible to attain the minimax rate $\Omega\left(\frac{\Delta L}{\epsilon^2} + \frac{\Delta L \sigma^{\frac{p}{p-1}}}{\epsilon^{\frac{3p-2}{p-1}}}\right)$ (Zhang et al., 2020b; Liu & Zhou, 2025; Liu, 2026) for non-convex optimization under heavy-tailed noise. Moreover, even if $\Delta$ and $L$ are known, Theorem 3.3 indicates that an extra polylogarithmic factor is also unavoidable for AdaGrad. These two facts together reveal some fundamental limitations of AdaGrad.

Since the proof is rather technical, we provide only a brief overview here and refer the interested reader to Appendix B for the analysis. Our proof builds upon the framework developed in Jiang et al. (2025) and Hübler et al. (2025). Concretely, we show that for a certain stochastic gradient oracle, one can construct a function parameterized by the learning rate $\gamma$ such that, with constant probability, $f'(x_t) \geq \Omega(\epsilon)$ for any $t \in [T]$ if $T$ is smaller than a threshold that also depends on $\gamma$. As a consequence, we derive an algorithm-dependent lower bound that explicitly captures the dependence on the input learning rate.

Finally, we suspect that our lower bound is not tight in $\epsilon$. Improving it could be an interesting task that we hope will be addressed in the future.

## 4. AdaGrad-Norm Can be Faster, Conditionally

In this section, we consider a variant of AdaGrad, namely AdaGrad-Norm (Algorithm 2). Unlike AdaGrad, AdaGrad-Norm no longer maintains a coordinate-wise update rule. Instead, its stepsize is now a scalar and is constructed based on the accumulated squared norm of the stochastic gradients in history. Although AdaGrad-Norm is not widely implemented in practice, it is popular in theoretical studies due to its simplicity.

*Remark* 4.1. Our analysis of AdaGrad-Norm still applies under the common norm-based heavy-tailed assumption, i.e., $\mathbb{E}_{t-1}\left[\|\xi_t\|_2^p\right] \leq \sigma^p$ for some $\sigma \geq 0$. However, to avoid introducing more assumptions, we still consider the coordinate-wise Assumption 2.4 in the analysis of AdaGrad-Norm.

### 4.1. A Faster Upper Bound

**Theorem 4.2.** *Under Assumptions 2.1, 2.2, 2.3, and 2.4, suppose $f^\star \triangleq \sup_{x \in \mathbb{R}^d} f(x) < +\infty$, let $\Delta_\star \triangleq f^\star - f_\star$, then for any $\gamma > 0$ and $\lambda > 0$, AdaGrad-Norm (Algorithm 2) guarantees*

$$\mathbb{E}\left[\frac{1}{T}\sum_{t=1}^{T}\|\nabla f(x_t)\|_2\right] \leq \mathcal{O}\left(\frac{A}{\sqrt{T}} + \frac{\sqrt{B\|\sigma\|_p}}{T^{\frac{p-1}{2p}}}\right),$$

*where $A \triangleq \sqrt{\frac{\lambda \Delta_\star}{\gamma}} + \frac{\Delta_\star}{\gamma} + \gamma\|L\|_\infty$ and $B \triangleq \frac{\Delta_\star}{\gamma} + \gamma\|L\|_\infty$.*

*Remark* 4.3. In fact, we can prove a bound using $\mathbb{E}\left[\frac{1}{T}\sum_{t=1}^{T}\|\nabla f(x_t)\|_2^2\right]$ as a stricter convergence metric (see (5) later).

*Remark* 4.4. Without additionally assuming an upper boundedness on the objective function, we can still show that AdaGrad-Norm converges at a rate of $\tilde{\mathcal{O}}(1/T^{\frac{3p-4}{4p}})$, in the same order as Theorem 3.1. The interested reader could refer to Theorem C.1 in Appendix C for details.

The key result in this section is Theorem 4.2, stating a faster rate for AdaGrad-Norm under an extra assumption of bounded objectives, which has also been considered in the existing literature on adaptive gradient methods (e.g., Levy et al. (2021); Chezhegov et al. (2025)).

Before moving on, we would like to discuss Theorem 4.2 further. First, it gives a faster rate $\mathcal{O}(1/T^{\frac{p-1}{2p}})$ without any extra polylogarithmic terms, which never becomes vacuous for any $p \in (1, 2]$, significantly improving upon Theorem 3.1. Moreover, similar to Theorem 3.1, it still adapts to the tail index $p$ and the noise level $\|\sigma\|_p$[3], reflecting the power of adaptive gradient methods. More interestingly, this result perfectly matches the best-known rate $\mathcal{O}(1/T^{\frac{p-1}{2p}})$ achieved by NSGD(M) in the case where problem-dependent parameters are unknown in advance (Hübler et al., 2025; Liu & Zhou, 2025).

Finally, we would like to comment that, even under the assumption of bounded objective functions, it is unclear to us whether AdaGrad can achieve the rate $\mathcal{O}(1/T^{\frac{p-1}{2p}})$, since

---

[3]Note that the change from $\|\sigma\|_1$ in Theorem 3.1 to $\|\sigma\|_p$ in Theorem 4.2 is reasonable and can be expected, since we are using the 2-norm as the convergence metric.

the proof strategy of Theorem 4.2 is specifically designed for `AdaGrad-Norm` (see Remark 4.5).

## 4.2. Theoretical Analysis

In this subsection, we aim to prove Theorem 4.2. Our proof is strongly inspired by the recent work of Liu (2026), which shows that `AdaGrad-Norm` guarantees an optimal regret bound in online convex optimization under heavy-tailed noise, even without knowing $p$. Although the problems studied are quite different, one will see that the underlying ideas and proof strategies are closely related (see Remark 4.6).

*Proof of Theorem 4.2.* In the following proof, let us denote the stepsize in `AdaGrad-Norm` by $\gamma_t \triangleq \frac{\gamma}{\lambda + \sqrt{v_t}}, \forall t \in \mathbb{N}$.

We start with Assumption 2.2 and use the update rule of `AdaGrad-Norm` to obtain

$$f(\boldsymbol{x}_{t+1}) \leq f(\boldsymbol{x}_t) + \langle \nabla f(\boldsymbol{x}_t), \boldsymbol{x}_{t+1} - \boldsymbol{x}_t \rangle + \frac{\|\boldsymbol{x}_{t+1} - \boldsymbol{x}_t\|_{\boldsymbol{L}}^2}{2}$$

$$= f(\boldsymbol{x}_t) - \gamma_t \langle \nabla f(\boldsymbol{x}_t), \boldsymbol{g}_t \rangle + \frac{\gamma_t^2 \|\boldsymbol{g}_t\|_{\boldsymbol{L}}^2}{2}$$

$$\leq f(\boldsymbol{x}_t) - \gamma_t \langle \nabla f(\boldsymbol{x}_t), \boldsymbol{g}_t \rangle + \frac{\gamma_t^2 \|\boldsymbol{L}\|_\infty \|\boldsymbol{g}_t\|_2^2}{2}.$$

Divide both sides of the above inequality by $\gamma_t$ and rearrange terms to have

$$\langle \nabla f(\boldsymbol{x}_t), \boldsymbol{g}_t \rangle \leq \frac{f(\boldsymbol{x}_t) - f(\boldsymbol{x}_{t+1})}{\gamma_t} + \frac{\gamma_t \|\boldsymbol{L}\|_\infty \|\boldsymbol{g}_t\|_2^2}{2},$$

which implies the following inequality after taking expectations on both sides (due to Assumption 2.3),

$$\mathbb{E}\left[\|\nabla f(\boldsymbol{x}_t)\|_2^2\right]$$

$$\leq \mathbb{E}\left[\frac{f(\boldsymbol{x}_t) - f(\boldsymbol{x}_{t+1})}{\gamma_t}\right] + \frac{\|\boldsymbol{L}\|_\infty \mathbb{E}\left[\gamma_t \|\boldsymbol{g}_t\|_2^2\right]}{2}. \quad (1)$$

*Remark* 4.5. The above derivation cannot be applied to `AdaGrad` due to the coordinate-wise stepsize in it.

*Remark* 4.6. The reader familiar with the online convex optimization literature, and in particular with the regret analysis of `AdaGrad-Norm`, may notice that (1) is closely related to the standard inequality characterizing single-step progress, if recognizing $f(\boldsymbol{x}_t) - f_\star$ as a notion of "distance". Therefore, under the boundedness assumption on the objective function, one may expect that $\mathbb{E}\left[\sum_{t=1}^T \|\nabla f(\boldsymbol{x}_t)\|_2^2\right]$ grows sublinearly in $T$.

However, the classical regret bound for `AdaGrad-Norm` is proved in the deterministic setting (or under finite-variance noise). Thanks to recent progress by Liu (2026), which has proved that `AdaGrad-Norm` is also robust to heavy-tailed

noise and achieves an optimal regret bound without knowing $p$. Inspired by Liu (2026), we are able to present the following analysis.

We sum (1) from $t = 1$ to $T$ to have

$$\mathbb{E}\left[\sum_{t=1}^T \|\nabla f(\boldsymbol{x}_t)\|_2^2\right]$$

$$\leq \mathbb{E}\left[\sum_{t=1}^T \frac{f(\boldsymbol{x}_t) - f(\boldsymbol{x}_{t+1})}{\gamma_t}\right] + \frac{\|\boldsymbol{L}\|_\infty}{2} \mathbb{E}\left[\sum_{t=1}^T \gamma_t \|\boldsymbol{g}_t\|_2^2\right]. \quad (2)$$

Observe that

$$\sum_{t=1}^T \frac{f(\boldsymbol{x}_t) - f(\boldsymbol{x}_{t+1})}{\gamma_t}$$

$$= \frac{f(\boldsymbol{x}_1) - f_\star}{\gamma_1} - \frac{f(\boldsymbol{x}_{T+1}) - f_\star}{\gamma_T}$$

$$+ \sum_{t=1}^{T-1} \left(\frac{1}{\gamma_{t+1}} - \frac{1}{\gamma_t}\right) (f(\boldsymbol{x}_{t+1}) - f_\star)$$

$$\leq \frac{f(\boldsymbol{x}_1) - f_\star}{\gamma_1} + \sum_{t=1}^{T-1} \left(\frac{1}{\gamma_{t+1}} - \frac{1}{\gamma_t}\right) (f(\boldsymbol{x}_{t+1}) - f_\star)$$

$$\overset{(a)}{\leq} \frac{\Delta_\star}{\gamma_T} = \frac{\lambda \Delta_\star}{\gamma} + \frac{\Delta_\star}{\gamma}\sqrt{v_T}, \quad (3)$$

where $(a)$ is due to

$$\frac{1}{\gamma_t} \leq \frac{1}{\gamma_{t+1}}, \quad f \leq f^\star, \quad \Delta_\star = f^\star - f_\star.$$

Moreover, we note that

$$\gamma_t \|\boldsymbol{g}_t\|_2^2 = \frac{\gamma \|\boldsymbol{g}_t\|_2^2}{\lambda + \sqrt{v_t}} \leq \frac{\gamma \|\boldsymbol{g}_t\|_2^2}{\sqrt{\lambda^2 + v_t}}$$

$$\leq 2\gamma \left(\sqrt{\lambda^2 + v_t} - \sqrt{\lambda^2 + v_{t-1}}\right)$$

$$\Rightarrow \sum_{t=1}^T \gamma_t \|\boldsymbol{g}_t\|_2^2 \leq 2\gamma \left(\sqrt{\lambda^2 + v_T} - \lambda\right) \leq 2\gamma\sqrt{v_T}. \quad (4)$$

To ease the notation, we write $u_T = \sum_{t=1}^T \|\nabla f(\boldsymbol{x}_t)\|_2^2$. Plug (3) and (4) back into (2) to obtain

$$\mathbb{E}[u_T] \leq \frac{\lambda \Delta_\star}{\gamma} + \left(\frac{\Delta_\star}{\gamma} + \gamma \|\boldsymbol{L}\|_\infty\right) \mathbb{E}[\sqrt{v_T}]$$

$$\overset{(b)}{\leq} \frac{\lambda \Delta_\star}{\gamma} + \sqrt{2} \left(\frac{\Delta_\star}{\gamma} + \gamma \|\boldsymbol{L}\|_\infty\right) \left(\|\boldsymbol{\sigma}\|_p T^{\frac{1}{p}} + \mathbb{E}[\sqrt{u_T}]\right)$$

$$\overset{(c)}{\leq} \frac{\lambda \Delta_\star}{\gamma} + \sqrt{2} \left(\frac{\Delta_\star}{\gamma} + \gamma \|\boldsymbol{L}\|_\infty\right) \left(\|\boldsymbol{\sigma}\|_p T^{\frac{1}{p}} + \sqrt{\mathbb{E}[u_T]}\right),$$

where $(b)$ holds by Lemma 4.7 and $(c)$ is due to Hölder's inequality. Note that by AM-GM inequality

$$\sqrt{2}\left(\frac{\Delta_\star}{\gamma} + \gamma \|\boldsymbol{L}\|_\infty\right)\sqrt{\mathbb{E}\left[u_T\right]}$$
$$\leq \left(\frac{\Delta_\star}{\gamma} + \gamma \|\boldsymbol{L}\|_\infty\right)^2 + \frac{\mathbb{E}\left[u_T\right]}{2}.$$

Next, we rearrange terms, plug in $u_T = \sum_{t=1}^{T} \|\nabla f(\boldsymbol{x}_t)\|_2^2$, and divide both sides by $T$ to obtain

$$\mathbb{E}\left[\frac{1}{T}\sum_{t=1}^{T}\|\nabla f(\boldsymbol{x}_t)\|_2^2\right] \leq \mathcal{O}\left(\frac{A^2}{T} + \frac{B\|\boldsymbol{\sigma}\|_p}{T^{\frac{p-1}{p}}}\right), \quad (5)$$

where $A = \sqrt{\frac{\lambda\Delta_\star}{\gamma}} + \frac{\Delta_\star}{\gamma} + \gamma\|\boldsymbol{L}\|_\infty$ and $B = \frac{\Delta_\star}{\gamma} + \gamma\|\boldsymbol{L}\|_\infty$ are defined in the statement of Theorem 4.2.

Finally, we can apply the following inequality to recover Theorem 4.2,

$$\mathbb{E}\left[\frac{1}{T}\sum_{t=1}^{T}\|\nabla f(\boldsymbol{x}_t)\|_2\right] \leq \mathbb{E}\left[\sqrt{\frac{1}{T}\sum_{t=1}^{T}\|\nabla f(\boldsymbol{x}_t)\|_2^2}\right]$$
$$\leq \sqrt{\mathbb{E}\left[\frac{1}{T}\sum_{t=1}^{T}\|\nabla f(\boldsymbol{x}_t)\|_2^2\right]} \overset{(5)}{\leq} \mathcal{O}\left(\frac{A}{\sqrt{T}} + \frac{\sqrt{B\|\boldsymbol{\sigma}\|_p}}{T^{\frac{p-1}{2p}}}\right),$$

where the first step holds by the concavity of $\sqrt{x}$ and the second step is due to Hölder's inequality. $\square$

The above proof relies on the following lemma, which shows that the term $\mathbb{E}\left[\sqrt{v_t}\right]$ can be upper bounded by $\|\boldsymbol{\sigma}\|_p t^{\frac{1}{p}}$ and $\mathbb{E}\left[\sqrt{u_t}\right]$ for any $p \in [1,2]$. Essentially the same observation was also made in Liu (2026).

**Lemma 4.7.** *Under Assumption 2.4, for any* $t \in \mathbb{N}$, AdaGrad-Norm *(Algorithm 2) guarantees*

$$\mathbb{E}\left[\sqrt{v_t}\right] \leq \sqrt{2}\|\boldsymbol{\sigma}\|_p t^{\frac{1}{p}} + \mathbb{E}\left[\sqrt{2u_t}\right],$$

*where* $u_t \triangleq \sum_{s=1}^{t}\|\nabla f(\boldsymbol{x}_s)\|_2^2, \forall t \in \mathbb{N}$.

*Proof.* By the definition of $v_t$, we have

$$\sqrt{v_t} = \sqrt{\sum_{s=1}^{t}\|\boldsymbol{g}_s\|_2^2} = \sqrt{\sum_{s=1}^{t}\|\boldsymbol{\xi}_s + \nabla f(\boldsymbol{x}_s)\|_2^2}$$
$$\leq \sqrt{2\sum_{s=1}^{t}\|\boldsymbol{\xi}_s\|_2^2 + 2\sum_{s=1}^{t}\|\nabla f(\boldsymbol{x}_s)\|_2^2}$$
$$\leq \sqrt{2\sum_{s=1}^{t}\|\boldsymbol{\xi}_s\|_2^2 + \sqrt{2u_t}}$$
$$\leq \sqrt{2}\left(\sum_{s=1}^{t}\sum_{i=1}^{d}|\boldsymbol{\xi}_{s,i}|^p\right)^{\frac{1}{p}} + \sqrt{2u_t},$$

where the last step is by applying $\|\cdot\|_2 \leq \|\cdot\|_p$ twice when $p \in [1,2]$, i.e.,

$$\sqrt{\sum_{s=1}^{t}\|\boldsymbol{\xi}_s\|_2^2} \leq \sqrt{\sum_{s=1}^{t}\|\boldsymbol{\xi}_s\|_p^2} \leq \left(\sum_{s=1}^{t}\|\boldsymbol{\xi}_s\|_p^p\right)^{\frac{1}{p}}.$$

Finally, by Hölder's inequality, we conclude that

$$\mathbb{E}\left[\sqrt{v_t}\right] \leq \sqrt{2}\left(\mathbb{E}\left[\sum_{s=1}^{t}\sum_{i=1}^{d}|\boldsymbol{\xi}_{s,i}|^p\right]\right)^{\frac{1}{p}} + \mathbb{E}\left[\sqrt{2u_t}\right]$$
$$\leq \sqrt{2}\|\boldsymbol{\sigma}\|_p t^{\frac{1}{p}} + \mathbb{E}\left[\sqrt{2u_t}\right],$$

where the last step is due to $\mathbb{E}\left[|\boldsymbol{\xi}_{s,i}|^p\right] \leq \boldsymbol{\sigma}_i^p, \forall s \in [t], i \in [d]$ by Assumption 2.4. $\square$

## 5. Conclusion, Limitations, and Future Work

**Conclusion.** This work makes the first attempt to understand whether AdaGrad, the origin of adaptive gradient methods, can converge in non-convex optimization under heavy-tailed noise. We partially address this question by establishing the first convergence rate for AdaGrad when the tail index $p$ lies in the range $(4/3, 2]$. Importantly, the obtained rate adapts to the tail index and the noise level simultaneously. Moreover, we derive an algorithm-dependent lower bound for AdaGrad in the same setting when $d = 1$, suggesting that the existing minimax rate for heavy-tailed non-convex optimization is not attainable by AdaGrad. In addition, we show that AdaGrad-Norm, a popular variant of AdaGrad in theoretical studies, can achieve a faster rate for all $p \in (1,2]$ when the objective function is bounded. We believe these results shed new light on the empirical success of adaptive gradient methods.

**Limitations.** This study has two main limitations. First, the derived upper bound for AdaGrad becomes vacuous when $p \in (1, 4/3]$. Second, the algorithm-dependent lower bound does not significantly improve upon the minimax rate in terms of $\epsilon$. Determining whether these upper and lower bounds are both loose, or whether one of them is in fact tight, is an important direction for future research.

**Future Work.** Several promising directions remain open for future investigation. For example, it is worthwhile to study whether more widely implemented adaptive gradient methods (e.g., Adam and AdamW) can converge under heavy-tailed noise and to characterize their limitations, thereby helping to demystify their strong performance in practice. Another direction is to analyze the convergence behavior of adaptive gradient methods under both heavy-tailed noise and other more realistic assumptions, such as the generalized smoothness condition proposed by Zhang et al. (2020a).

## Acknowledgements

The author thanks the anonymous reviewers for their valuable feedback.

## Impact Statement

This paper presents work whose goal is to advance the field of machine learning. There are many potential societal consequences of our work, none of which we feel must be specifically highlighted here.

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

# A. Upper Bound for `AdaGrad`

This section provides the full statement of Theorem 3.1 and its proof.

## A.1. Full Theorem and Its Proof

**Theorem A.1** (Full statement of Theorem 3.1). *Under Assumptions 2.1, 2.2, 2.3, and 2.4, let $\Delta \triangleq f(\boldsymbol{x}_1) - f_\star$, then for any $\gamma > 0$ and $\lambda > 0$,* `AdaGrad` *(Algorithm 1) guarantees*

$$\mathbb{E}\left[\frac{1}{T}\sum_{t=1}^{T}\|\nabla f(\boldsymbol{x}_t)\|_1\right] \leq \mathcal{O}\left(\frac{d\lambda + \frac{\Delta}{\gamma} + \gamma\|\boldsymbol{L}\|_1 \ln K_T}{\sqrt{T}} + \frac{\|\boldsymbol{\sigma}\|_1 \ln^{\frac{1}{p}+\frac{1}{2}} K_T}{T^{\frac{p-1}{p}}}\right.$$

$$\left. + \frac{\sqrt{\left(\frac{\Delta}{\gamma} + \gamma\|\boldsymbol{L}\|_1 \ln K_T\right)\|\boldsymbol{\sigma}\|_1}}{T^{\frac{p-1}{2p}}} + \frac{\|\boldsymbol{\sigma}\|_1 \ln^{\frac{1}{2p}+\frac{1}{4}} K_T}{T^{\frac{3p-4}{4p}}}\right),$$

*where $K_T = 1 + \frac{\sqrt{2}\|\boldsymbol{\sigma}\|_\infty T^{\frac{1}{p}} + 2\|\nabla f(\boldsymbol{x}_1)\|_\infty T^{\frac{1}{2}} + 2\gamma\sqrt{\|\boldsymbol{L}\|_1\|\boldsymbol{L}\|_\infty} T^{\frac{3}{2}}}{\lambda}$ is introduced in Lemma A.5.*

*Proof.* In the following proof, let

$$\boldsymbol{c}_i \triangleq \frac{\boldsymbol{\sigma}_i T^{\frac{1}{2}-\frac{1}{p}}}{D_{T,i}^{\frac{1}{2}-\frac{1}{p}}}, \forall i \in [d], \tag{6}$$

where

$$D_{T,i} \triangleq 2\ln\left(1 + \frac{\sqrt{2}\boldsymbol{\sigma}_i T^{\frac{1}{p}} + \mathbb{E}\left[\sqrt{2\boldsymbol{u}_{T,i}}\right]}{\lambda}\right), \forall i \in [d]. \tag{7}$$

By Lemma A.5, we have

$$\mathbb{E}\left[\ln\left(1 + \frac{\boldsymbol{v}_{T,i}}{\lambda^2}\right)\right] \leq D_{T,i} \leq 2\ln K_T. \tag{8}$$

We sum up the inequality in Lemma A.2 (with $\boldsymbol{c}$ defined in (6)) from $t = 1$ to $T$ and use $f(\boldsymbol{x}_1) - f(\boldsymbol{x}_{T+1}) \leq \Delta$ (Assumption 2.1) to have

$$\frac{\gamma}{2}\mathbb{E}\left[\sum_{t=1}^{T}\left\|\frac{(\nabla f(\boldsymbol{x}_t))^2}{\lambda + \sqrt{\boldsymbol{w}_t}}\right\|_1\right] \leq \Delta + \gamma\sum_{i=1}^{d}\frac{\boldsymbol{\sigma}_i^2}{\boldsymbol{c}_i}\sum_{t=1}^{T}\left(\mathbb{E}\left[\frac{\boldsymbol{g}_{t,i}^2}{\lambda^2 + \boldsymbol{v}_{t,i}}\right]\right)^{\frac{2}{p}} + \gamma\sum_{i=1}^{d}\left(\boldsymbol{c}_i + \frac{\gamma\boldsymbol{L}_i}{2}\right)\sum_{t=1}^{T}\mathbb{E}\left[\frac{\boldsymbol{g}_{t,i}^2}{\lambda^2 + \boldsymbol{v}_{t,i}}\right]$$

$$\stackrel{(a)}{\leq} \Delta + \gamma\sum_{i=1}^{d}\frac{\boldsymbol{\sigma}_i^2}{\boldsymbol{c}_i}T^{1-\frac{2}{p}}\left(\mathbb{E}\left[\ln\left(1 + \frac{\boldsymbol{v}_{T,i}}{\lambda^2}\right)\right]\right)^{\frac{2}{p}} + \gamma\sum_{i=1}^{d}\left(\boldsymbol{c}_i + \frac{\gamma\boldsymbol{L}_i}{2}\right)\mathbb{E}\left[\ln\left(1 + \frac{\boldsymbol{v}_{T,i}}{\lambda^2}\right)\right]$$

$$\stackrel{(8)}{\leq} \Delta + \gamma\sum_{i=1}^{d}\left(\frac{\boldsymbol{\sigma}_i^2}{\boldsymbol{c}_i}T^{1-\frac{2}{p}}D_{T,i}^{\frac{2}{p}} + \boldsymbol{c}_i D_{T,i}\right) + \gamma^2\|\boldsymbol{L}\|_1 \ln K_T$$

$$\stackrel{(6)}{=} \Delta + 2\gamma\sum_{i=1}^{d}\boldsymbol{\sigma}_i T^{\frac{1}{2}-\frac{1}{p}}D_{T,i}^{\frac{1}{p}+\frac{1}{2}} + \gamma^2\|\boldsymbol{L}\|_1 \ln K_T$$

$$\stackrel{(8)}{\leq} \Delta + 2^{\frac{1}{p}+\frac{3}{2}}\gamma\|\boldsymbol{\sigma}\|_1 T^{\frac{1}{2}-\frac{1}{p}}\ln^{\frac{1}{p}+\frac{1}{2}} K_T + \gamma^2\|\boldsymbol{L}\|_1 \ln K_T$$

$$\stackrel{(b)}{=} \Delta + 4\gamma\|\boldsymbol{\sigma}\|_1 T^{\frac{1}{p}-\frac{1}{2}}\ln^{\frac{1}{p}+\frac{1}{2}} K_T + \gamma^2\|\boldsymbol{L}\|_1 \ln K_T,$$

where $(a)$ is by applying Lemma A.3 with $q = \frac{2}{p}$ and $q = 1$ and $(b)$ is due to $\frac{1}{p} \leq \frac{1}{2}$ and $\frac{1}{2} - \frac{1}{p} = \frac{1}{p} - \frac{1}{2}$. Divide both sides of the above inequality by $\frac{\gamma}{2}$ to obtain

$$\mathbb{E}\left[\sum_{t=1}^{T}\left\|\frac{(\nabla f(\boldsymbol{x}_t))^2}{\lambda + \sqrt{\boldsymbol{w}_t}}\right\|_1\right] \leq \frac{2\Delta}{\gamma} + 2\gamma\|\boldsymbol{L}\|_1 \ln K_T + 8\|\boldsymbol{\sigma}\|_1 T^{\frac{1}{p}-\frac{1}{2}}\ln^{\frac{1}{p}+\frac{1}{2}} K_T. \tag{9}$$

Next, recall that $\boldsymbol{u}_{t,i} = \sum_{s=1}^{t}(\nabla_i f(\boldsymbol{x}_s))^2, \forall t \in \mathbb{N}$ (introduced in Lemma A.4), we hence have

$$
\begin{aligned}
\mathbb{E}\left[\sqrt{\boldsymbol{u}_{T,i}}\right] &= \mathbb{E}\left[\sqrt{\frac{\boldsymbol{u}_{T,i}}{\lambda + \sqrt{\boldsymbol{v}_{T,i} + \boldsymbol{u}_{T,i} + \boldsymbol{c}_i^2}} \times \left(\lambda + \sqrt{\boldsymbol{v}_{T,i} + \boldsymbol{u}_{T,i} + \boldsymbol{c}_i^2}\right)}\right] \\
&\overset{(c)}{\leq} \sqrt{\mathbb{E}\left[\frac{\boldsymbol{u}_{T,i}}{\lambda + \sqrt{\boldsymbol{v}_{T,i} + \boldsymbol{u}_{T,i} + \boldsymbol{c}_i^2}}\right]\mathbb{E}\left[\lambda + \sqrt{\boldsymbol{v}_{T,i} + \boldsymbol{u}_{T,i} + \boldsymbol{c}_i^2}\right]} \\
&\leq \sqrt{\mathbb{E}\left[\sum_{t=1}^{T}\frac{(\nabla_i f(\boldsymbol{x}_t))^2}{\lambda + \sqrt{\boldsymbol{w}_{t,i}}}\right]\mathbb{E}\left[\lambda + \sqrt{\boldsymbol{v}_{T,i} + \boldsymbol{u}_{T,i} + \boldsymbol{c}_i^2}\right]} \\
&\overset{(d)}{\leq} \sqrt{\mathbb{E}\left[\sum_{t=1}^{T}\frac{(\nabla_i f(\boldsymbol{x}_t))^2}{\lambda + \sqrt{\boldsymbol{w}_{t,i}}}\right]\left(\lambda + \boldsymbol{c}_i + \sqrt{2}\boldsymbol{\sigma}_i T^{\frac{1}{p}} + \sqrt{3}\mathbb{E}\left[\sqrt{\boldsymbol{u}_{T,i}}\right]\right)} \\
&\overset{(6)}{=} \sqrt{\mathbb{E}\left[\sum_{t=1}^{T}\frac{(\nabla_i f(\boldsymbol{x}_t))^2}{\lambda + \sqrt{\boldsymbol{w}_{t,i}}}\right]\left(\lambda + \frac{\boldsymbol{\sigma}_i}{D_{T,i}^{\frac{1}{2}-\frac{1}{p}}}T^{\frac{1}{2}-\frac{1}{p}} + \sqrt{2}\boldsymbol{\sigma}_i T^{\frac{1}{p}} + \sqrt{3}\mathbb{E}\left[\sqrt{\boldsymbol{u}_{T,i}}\right]\right)},
\end{aligned}
\tag{10}
$$

where $(c)$ is by Hölder's inequality and $(d)$ follows similar steps to proving Lemma A.4. Now, we consider two cases:

*Case* 1.   $D_{T,i} \geq 1$: in this case, we have

$$
\frac{\boldsymbol{\sigma}_i}{D_{T,i}^{\frac{1}{2}-\frac{1}{p}}}T^{\frac{1}{2}-\frac{1}{p}} \leq \boldsymbol{\sigma}_i T^{\frac{1}{2}-\frac{1}{p}} = \boldsymbol{\sigma}_i T^{\frac{1}{p}-\frac{1}{2}} \leq \boldsymbol{\sigma}_i T^{\frac{1}{p}},
$$

which implies that

$$
\mathbb{E}\left[\sqrt{\boldsymbol{u}_{T,i}}\right] \overset{(10)}{\leq} \sqrt{\mathbb{E}\left[\sum_{t=1}^{T}\frac{(\nabla_i f(\boldsymbol{x}_t))^2}{\lambda + \sqrt{\boldsymbol{w}_{t,i}}}\right]\left(\lambda + \left(1 + \sqrt{2}\right)\boldsymbol{\sigma}_i T^{\frac{1}{p}} + \sqrt{3}\mathbb{E}\left[\sqrt{\boldsymbol{u}_{T,i}}\right]\right)}.
$$

*Case* 2.   $D_{T,i} < 1$: in this case, we have

$$
1 > D_{T,i} \overset{(7)}{=} 2\ln\left(1 + \frac{\sqrt{2}\boldsymbol{\sigma}_i T^{\frac{1}{p}} + \mathbb{E}\left[\sqrt{2\boldsymbol{u}_{T,i}}\right]}{\lambda}\right),
$$

which implies that

$$
\mathbb{E}\left[\sqrt{\boldsymbol{u}_{T,i}}\right] \leq \boldsymbol{\sigma}_i T^{\frac{1}{p}} + \mathbb{E}\left[\sqrt{\boldsymbol{u}_{T,i}}\right] \leq \frac{\sqrt{e}-1}{\sqrt{2}}\lambda.
$$

Therefore, we always have

$$
\mathbb{E}\left[\sqrt{\boldsymbol{u}_{T,i}}\right] \leq \sqrt{\mathbb{E}\left[\sum_{t=1}^{T}\frac{(\nabla_i f(\boldsymbol{x}_t))^2}{\lambda + \sqrt{\boldsymbol{w}_{t,i}}}\right]\left(\lambda + \left(1 + \sqrt{2}\right)\boldsymbol{\sigma}_i T^{\frac{1}{p}} + \sqrt{3}\mathbb{E}\left[\sqrt{\boldsymbol{u}_{T,i}}\right]\right)} + \frac{\sqrt{e}-1}{\sqrt{2}}\lambda.
$$

Sum up the above inequality for all $i \in [d]$ to have

$$
\mathbb{E}\left[\sum_{i=1}^{d}\sqrt{\boldsymbol{u}_{T,i}}\right] \leq \sum_{i=1}^{d}\sqrt{\mathbb{E}\left[\sum_{t=1}^{T}\frac{(\nabla_i f(\boldsymbol{x}_t))^2}{\lambda+\sqrt{\boldsymbol{w}_{t,i}}}\right]\left(\lambda+\left(1+\sqrt{2}\right)\boldsymbol{\sigma}_i T^{\frac{1}{p}}+\sqrt{3}\mathbb{E}\left[\sqrt{\boldsymbol{u}_{T,i}}\right]\right)+\frac{\sqrt{e}-1}{\sqrt{2}}d\lambda}
$$

$$
\overset{(e)}{\leq}\sqrt{\left(\sum_{i=1}^{d}\mathbb{E}\left[\sum_{t=1}^{T}\frac{(\nabla_i f(\boldsymbol{x}_t))^2}{\lambda+\sqrt{\boldsymbol{w}_{t,i}}}\right]\right)\left(d\lambda+\left(1+\sqrt{2}\right)\|\boldsymbol{\sigma}\|_1 T^{\frac{1}{p}}+\sqrt{3}\mathbb{E}\left[\sum_{i=1}^{d}\sqrt{\boldsymbol{u}_{T,i}}\right]\right)+\frac{\sqrt{e}-1}{\sqrt{2}}d\lambda}
$$

$$
=\sqrt{\mathbb{E}\left[\sum_{t=1}^{T}\left\|\frac{(\nabla f(\boldsymbol{x}_t))^2}{\lambda+\sqrt{\boldsymbol{w}_t}}\right\|_1\right]\left(d\lambda+\left(1+\sqrt{2}\right)\|\boldsymbol{\sigma}\|_1 T^{\frac{1}{p}}+\sqrt{3}\mathbb{E}\left[\sum_{i=1}^{d}\sqrt{\boldsymbol{u}_{T,i}}\right]\right)+\frac{\sqrt{e}-1}{\sqrt{2}}d\lambda}
$$

$$
\Rightarrow \mathbb{E}\left[\sum_{i=1}^{d}\sqrt{\boldsymbol{u}_{T,i}}\right] \leq \mathcal{O}\left(d\lambda+\mathbb{E}\left[\sum_{t=1}^{T}\left\|\frac{(\nabla f(\boldsymbol{x}_t))^2}{\lambda+\sqrt{\boldsymbol{w}_t}}\right\|_1\right]+\sqrt{\mathbb{E}\left[\sum_{t=1}^{T}\left\|\frac{(\nabla f(\boldsymbol{x}_t))^2}{\lambda+\sqrt{\boldsymbol{w}_t}}\right\|_1\right]\|\boldsymbol{\sigma}\|_1 T^{\frac{1}{p}}}\right), \tag{11}
$$

where $(e)$ is due to Cauchy-Schwarz inequality.

Now, we combine (9) and (11) to obtain

$$
\mathbb{E}\left[\sum_{i=1}^{d}\sqrt{\boldsymbol{u}_{T,i}}\right] \leq \mathcal{O}\left(d\lambda+\frac{\Delta}{\gamma}+\gamma\|\boldsymbol{L}\|_1\ln K_T+\|\boldsymbol{\sigma}\|_1 T^{\frac{1}{p}-\frac{1}{2}}\ln^{\frac{1}{p}+\frac{1}{2}}K_T\right.
$$
$$
\left.+\sqrt{\left(\frac{\Delta}{\gamma}+\gamma\|\boldsymbol{L}\|_1\ln K_T+\|\boldsymbol{\sigma}\|_1 T^{\frac{1}{p}-\frac{1}{2}}\ln^{\frac{1}{p}+\frac{1}{2}}K_T\right)\|\boldsymbol{\sigma}\|_1 T^{\frac{1}{p}}}\right).
$$

Lastly, we observe that

$$
\sum_{i=1}^{d}\sqrt{\boldsymbol{u}_{T,i}}=\sum_{i=1}^{d}\sqrt{\sum_{t=1}^{T}(\nabla_i f(\boldsymbol{x}_t))^2} \geq \frac{\sum_{i=1}^{d}\sum_{t=1}^{T}|\nabla_i f(\boldsymbol{x}_t)|}{\sqrt{T}}=\frac{\sum_{t=1}^{T}\|\nabla f(\boldsymbol{x}_t)\|_1}{\sqrt{T}},
$$

which gives us

$$
\mathbb{E}\left[\frac{1}{T}\sum_{t=1}^{T}\|\nabla f(\boldsymbol{x}_t)\|_1\right] \leq \mathcal{O}\left(\frac{d\lambda+\frac{\Delta}{\gamma}+\gamma\|\boldsymbol{L}\|_1\ln K_T}{\sqrt{T}}+\frac{\|\boldsymbol{\sigma}\|_1\ln^{\frac{1}{p}+\frac{1}{2}}K_T}{T^{\frac{p-1}{p}}}\right.
$$
$$
\left.+\frac{\sqrt{\left(\frac{\Delta}{\gamma}+\gamma\|\boldsymbol{L}\|_1\ln K_T\right)\|\boldsymbol{\sigma}\|_1}}{T^{\frac{p-1}{2p}}}+\frac{\|\boldsymbol{\sigma}\|_1\ln^{\frac{1}{2p}+\frac{1}{4}}K_T}{T^{\frac{3p-4}{4p}}}\right).
$$

$\square$

## A.2. Helpful Lemmas

To prove Theorem A.1, we require the following four lemmas. Before presenting them, we recall a key ingredient in our analysis, the generalized proxy stepsize, defined as follows

$$
\boldsymbol{w}_t \triangleq \boldsymbol{v}_{t-1}+(\nabla f(\boldsymbol{x}_t))^2+\boldsymbol{c}^2 \in \mathcal{F}_{t-1},
$$

where $\boldsymbol{c} \in \mathbb{R}_{\geq 0}^d$ is a free parameter that will be specified in the final proof. As discussed in Section 3, it plays a crucial role in establishing the final convergence rate.

We are now ready to give the first result, Lemma A.2, which characterizes the per-iteration progress of `AdaGrad`.

**Lemma A.2.** *Under Assumptions 2.2, 2.3, and 2.4, for any $\boldsymbol{c} \in \mathbb{R}_{\geq 0}^d$ and $t \in \mathbb{N}$,* `AdaGrad` *(Algorithm 1) guarantees*

$$
\frac{\gamma}{2}\mathbb{E}\left[\left\|\frac{(\nabla f(\boldsymbol{x}_t))^2}{\lambda+\sqrt{\boldsymbol{w}_t}}\right\|_1\right] \leq \mathbb{E}\left[f(\boldsymbol{x}_t)-f(\boldsymbol{x}_{t+1})\right]+\gamma\sum_{i=1}^{d}\frac{\boldsymbol{\sigma}_i^2}{\boldsymbol{c}_i}\left(\mathbb{E}\left[\frac{\boldsymbol{g}_{t,i}^2}{\lambda^2+\boldsymbol{v}_{t,i}}\right]\right)^{\frac{2}{p}}+\gamma\sum_{i=1}^{d}\left(\boldsymbol{c}_i+\frac{\gamma\boldsymbol{L}_i}{2}\right)\mathbb{E}\left[\frac{\boldsymbol{g}_{t,i}^2}{\lambda^2+\boldsymbol{v}_{t,i}}\right].
$$

*Proof.* We start with Assumption 2.2 and use the update rule of `AdaGrad` to obtain

$$f(\boldsymbol{x}_{t+1}) \leq f(\boldsymbol{x}_t) + \langle \nabla f(\boldsymbol{x}_t), \boldsymbol{x}_{t+1} - \boldsymbol{x}_t \rangle + \frac{\|\boldsymbol{x}_{t+1} - \boldsymbol{x}_t\|_{\boldsymbol{L}}^2}{2}$$

$$= f(\boldsymbol{x}_t) - \gamma \left\langle \nabla f(\boldsymbol{x}_t), \frac{\boldsymbol{g}_t}{\lambda + \sqrt{\boldsymbol{v}_t}} \right\rangle + \frac{\gamma^2}{2} \left\| \frac{\boldsymbol{g}_t}{\lambda + \sqrt{\boldsymbol{v}_t}} \right\|_{\boldsymbol{L}}^2$$

$$\leq f(\boldsymbol{x}_t) - \gamma \left\langle \nabla f(\boldsymbol{x}_t), \frac{\boldsymbol{g}_t}{\lambda + \sqrt{\boldsymbol{v}_t}} \right\rangle + \frac{\gamma^2}{2} \left\| \frac{\boldsymbol{L}\boldsymbol{g}_t^2}{\lambda^2 + \boldsymbol{v}_t} \right\|_1$$

$$= f(\boldsymbol{x}_t) - \gamma \left\langle \nabla f(\boldsymbol{x}_t), \frac{\boldsymbol{g}_t}{\lambda + \sqrt{\boldsymbol{w}_t}} \right\rangle + \gamma \left\langle \nabla f(\boldsymbol{x}_t), \frac{\boldsymbol{g}_t}{\lambda + \sqrt{\boldsymbol{w}_t}} - \frac{\boldsymbol{g}_t}{\lambda + \sqrt{\boldsymbol{v}_t}} \right\rangle + \frac{\gamma^2}{2} \left\| \frac{\boldsymbol{L}\boldsymbol{g}_t^2}{\lambda^2 + \boldsymbol{v}_t} \right\|_1.$$

Take conditional expectations on both sides of the above inequality to obtain

$$\mathbb{E}_{t-1}\left[ f(\boldsymbol{x}_{t+1}) \right]$$

$$\leq f(\boldsymbol{x}_t) - \gamma \left\| \frac{(\nabla f(\boldsymbol{x}_t))^2}{\lambda + \sqrt{\boldsymbol{w}_t}} \right\|_1 + \gamma \mathbb{E}_{t-1}\left[ \left\langle \nabla f(\boldsymbol{x}_t), \frac{\boldsymbol{g}_t}{\lambda + \sqrt{\boldsymbol{w}_t}} - \frac{\boldsymbol{g}_t}{\lambda + \sqrt{\boldsymbol{v}_t}} \right\rangle \right] + \frac{\gamma^2}{2} \mathbb{E}_{t-1}\left[ \left\| \frac{\boldsymbol{L}\boldsymbol{g}_t^2}{\lambda^2 + \boldsymbol{v}_t} \right\|_1 \right]$$

$$= f(\boldsymbol{x}_t) - \gamma \left\| \frac{(\nabla f(\boldsymbol{x}_t))^2}{\lambda + \sqrt{\boldsymbol{w}_t}} \right\|_1 + \gamma \sum_{i=1}^d \mathbb{E}_{t-1}\left[ \frac{\nabla_i f(\boldsymbol{x}_t)\boldsymbol{g}_{t,i}}{\lambda + \sqrt{\boldsymbol{w}_{t,i}}} - \frac{\nabla_i f(\boldsymbol{x}_t)\boldsymbol{g}_{t,i}}{\lambda + \sqrt{\boldsymbol{v}_{t,i}}} \right] + \frac{\gamma^2}{2} \sum_{i=1}^d \mathbb{E}_{t-1}\left[ \frac{\boldsymbol{L}_i \boldsymbol{g}_{t,i}^2}{\lambda^2 + \boldsymbol{v}_{t,i}} \right], \quad (12)$$

where the first step is by

$$\mathbb{E}_{t-1}\left[ \left\langle \nabla f(\boldsymbol{x}_t), \frac{\boldsymbol{g}_t}{\lambda + \sqrt{\boldsymbol{w}_t}} \right\rangle \right] \stackrel{\boldsymbol{x}_t, \boldsymbol{w}_t \in \mathcal{F}_{t-1}}{=} \left\langle \nabla f(\boldsymbol{x}_t), \frac{\mathbb{E}_{t-1}[\boldsymbol{g}_t]}{\lambda + \sqrt{\boldsymbol{w}_t}} \right\rangle \stackrel{\text{Assumption 2.3}}{=} \left\langle \nabla f(\boldsymbol{x}_t), \frac{\nabla f(\boldsymbol{x}_t)}{\lambda + \sqrt{\boldsymbol{w}_t}} \right\rangle = \left\| \frac{(\nabla f(\boldsymbol{x}_t))^2}{\lambda + \sqrt{\boldsymbol{w}_t}} \right\|_1.$$

For any fixed coordinate $i \in [d]$, we can bound

$$\mathbb{E}_{t-1}\left[ \frac{\nabla_i f(\boldsymbol{x}_t)\boldsymbol{g}_{t,i}}{\lambda + \sqrt{\boldsymbol{w}_{t,i}}} - \frac{\nabla_i f(\boldsymbol{x}_t)\boldsymbol{g}_{t,i}}{\lambda + \sqrt{\boldsymbol{v}_{t,i}}} \right] = \mathbb{E}_{t-1}\left[ \frac{(\boldsymbol{g}_{t,i}^2 - (\nabla_i f(\boldsymbol{x}_t))^2 - \boldsymbol{c}_i^2)\nabla_i f(\boldsymbol{x}_t)\boldsymbol{g}_{t,i}}{(\lambda + \sqrt{\boldsymbol{w}_{t,i}})(\lambda + \sqrt{\boldsymbol{v}_{t,i}})(\sqrt{\boldsymbol{w}_{t,i}} + \sqrt{\boldsymbol{v}_{t,i}})} \right]$$

$$= \mathbb{E}_{t-1}\left[ \frac{(\boldsymbol{g}_{t,i} + \nabla_i f(\boldsymbol{x}_t))\boldsymbol{\xi}_{t,i}\nabla_i f(\boldsymbol{x}_t)\boldsymbol{g}_{t,i}}{(\lambda + \sqrt{\boldsymbol{w}_{t,i}})(\lambda + \sqrt{\boldsymbol{v}_{t,i}})(\sqrt{\boldsymbol{w}_{t,i}} + \sqrt{\boldsymbol{v}_{t,i}})} \right] + \mathbb{E}_{t-1}\left[ \frac{-\boldsymbol{c}_i^2 \nabla_i f(\boldsymbol{x}_t)\boldsymbol{g}_{t,i}}{(\lambda + \sqrt{\boldsymbol{w}_{t,i}})(\lambda + \sqrt{\boldsymbol{v}_{t,i}})(\sqrt{\boldsymbol{w}_{t,i}} + \sqrt{\boldsymbol{v}_{t,i}})} \right]$$

$$\leq \mathbb{E}_{t-1}\left[ \frac{(|\boldsymbol{g}_{t,i}| + |\nabla_i f(\boldsymbol{x}_t)|)|\boldsymbol{\xi}_{t,i}||\nabla_i f(\boldsymbol{x}_t)||\boldsymbol{g}_{t,i}|}{(\lambda + \sqrt{\boldsymbol{w}_{t,i}})(\lambda + \sqrt{\boldsymbol{v}_{t,i}})(\sqrt{\boldsymbol{w}_{t,i}} + \sqrt{\boldsymbol{v}_{t,i}})} \right] + \mathbb{E}_{t-1}\left[ \frac{\boldsymbol{c}_i^2 |\nabla_i f(\boldsymbol{x}_t)||\boldsymbol{g}_{t,i}|}{(\lambda + \sqrt{\boldsymbol{w}_{t,i}})(\lambda + \sqrt{\boldsymbol{v}_{t,i}})(\sqrt{\boldsymbol{w}_{t,i}} + \sqrt{\boldsymbol{v}_{t,i}})} \right]$$

$$\stackrel{(a)}{\leq} \mathbb{E}_{t-1}\left[ \frac{|\boldsymbol{\xi}_{t,i}||\nabla_i f(\boldsymbol{x}_t)||\boldsymbol{g}_{t,i}|}{(\lambda + \sqrt{\boldsymbol{w}_{t,i}})(\lambda + \sqrt{\boldsymbol{v}_{t,i}})} \right] + \mathbb{E}_{t-1}\left[ \frac{\boldsymbol{c}_i |\nabla_i f(\boldsymbol{x}_t)||\boldsymbol{g}_{t,i}|}{(\lambda + \sqrt{\boldsymbol{w}_{t,i}})(\lambda + \sqrt{\boldsymbol{v}_{t,i}})} \right]$$

$$\stackrel{(b)}{\leq} \frac{|\nabla_i f(\boldsymbol{x}_t)|}{\lambda + \sqrt{\boldsymbol{w}_{t,i}}} \left( \mathbb{E}_{t-1}\left[ \frac{|\boldsymbol{\xi}_{t,i}||\boldsymbol{g}_{t,i}|}{\lambda + \sqrt{\boldsymbol{v}_{t,i}}} \right] + \mathbb{E}_{t-1}\left[ \frac{\boldsymbol{c}_i |\boldsymbol{g}_{t,i}|}{\lambda + \sqrt{\boldsymbol{v}_{t,i}}} \right] \right)$$

$$\stackrel{(c)}{\leq} \frac{(\nabla_i f(\boldsymbol{x}_t))^2}{2(\lambda + \sqrt{\boldsymbol{w}_{t,i}})} + \frac{1}{\lambda + \sqrt{\boldsymbol{w}_{t,i}}} \left( \left( \mathbb{E}_{t-1}\left[ \frac{|\boldsymbol{\xi}_{t,i}||\boldsymbol{g}_{t,i}|}{\lambda + \sqrt{\boldsymbol{v}_{t,i}}} \right] \right)^2 + \left( \mathbb{E}_{t-1}\left[ \frac{\boldsymbol{c}_i |\boldsymbol{g}_{t,i}|}{\lambda + \sqrt{\boldsymbol{v}_{t,i}}} \right] \right)^2 \right), \quad (13)$$

where $(a)$ is due to $|\boldsymbol{g}_{t,i}| + |\nabla_i f(\boldsymbol{x}_t)| \leq \sqrt{\boldsymbol{w}_{t,i}} + \sqrt{\boldsymbol{v}_{t,i}}$ and $\boldsymbol{c}_i \leq \sqrt{\boldsymbol{w}_{t,i}} + \sqrt{\boldsymbol{v}_{t,i}}$, $(b)$ is from $\frac{|\nabla_i f(\boldsymbol{x}_t)|}{\lambda + \sqrt{\boldsymbol{w}_{t,i}}} \in \mathcal{F}_{t-1}$, and $(c)$ holds by AM-GM inequality, i.e., $|\nabla_i f(\boldsymbol{x}_t)| X \leq \frac{(\nabla_i f(\boldsymbol{x}_t))^2}{4} + X^2$ for $X = \mathbb{E}_{t-1}\left[ \frac{|\boldsymbol{\xi}_{t,i}||\boldsymbol{g}_{t,i}|}{\lambda + \sqrt{\boldsymbol{v}_{t,i}}} \right]$ and $\mathbb{E}_{t-1}\left[ \frac{\boldsymbol{c}_i |\boldsymbol{g}_{t,i}|}{\lambda + \sqrt{\boldsymbol{v}_{t,i}}} \right]$, respectively. Next, we apply Hölder's inequality to get

$$\left( \mathbb{E}_{t-1}\left[ \frac{|\boldsymbol{\xi}_{t,i}||\boldsymbol{g}_{t,i}|}{\lambda + \sqrt{\boldsymbol{v}_{t,i}}} \right] \right)^2 \leq \left( \mathbb{E}_{t-1}\left[ |\boldsymbol{\xi}_{t,i}|^p \right] \right)^{\frac{2}{p}} \left( \mathbb{E}_{t-1}\left[ \frac{|\boldsymbol{g}_{t,i}|^{\bar{p}}}{(\lambda + \sqrt{\boldsymbol{v}_{t,i}})^{\bar{p}}} \right] \right)^{\frac{2}{\bar{p}}}$$

$$\stackrel{\text{Assumption 2.4}}{\leq} \boldsymbol{\sigma}_i^2 \left( \mathbb{E}_{t-1}\left[ \frac{|\boldsymbol{g}_{t,i}|^{\bar{p}}}{(\lambda + \sqrt{\boldsymbol{v}_{t,i}})^{\bar{p}}} \right] \right)^{\frac{2}{\bar{p}}} \leq \boldsymbol{\sigma}_i^2 \left( \mathbb{E}_{t-1}\left[ \left( \frac{\boldsymbol{g}_{t,i}^2}{\lambda^2 + \boldsymbol{v}_{t,i}} \right)^{\frac{\bar{p}}{2}} \right] \right)^{\frac{2}{\bar{p}}}, \quad (14)$$

and

$$\left(\mathbb{E}_{t-1}\left[\frac{c_i\,|g_{t,i}|}{\lambda+\sqrt{v_{t,i}}}\right]\right)^2 \leq \mathbb{E}_{t-1}\left[\frac{c_i^2 g_{t,i}^2}{(\lambda+\sqrt{v_{t,i}})^2}\right] \leq \mathbb{E}_{t-1}\left[\frac{c_i^2 g_{t,i}^2}{\lambda^2+v_{t,i}}\right]. \tag{15}$$

Plug (14) and (15) into (13) to obtain

$$\mathbb{E}_{t-1}\left[\frac{\nabla_i f(x_t)g_{t,i}}{\lambda+\sqrt{w_{t,i}}} - \frac{\nabla_i f(x_t)g_{t,i}}{\lambda+\sqrt{v_{t,i}}}\right]$$

$$\leq \frac{(\nabla_i f(x_t))^2}{2(\lambda+\sqrt{w_{t,i}})} + \frac{\sigma_i^2}{\lambda+\sqrt{w_{t,i}}}\left(\mathbb{E}_{t-1}\left[\left(\frac{g_{t,i}^2}{\lambda^2+v_{t,i}}\right)^{\frac{\bar p}{2}}\right]\right)^{\frac{2}{\bar p}} + \frac{c_i^2}{\lambda+\sqrt{w_{t,i}}}\mathbb{E}_{t-1}\left[\frac{g_{t,i}^2}{\lambda^2+v_{t,i}}\right]$$

$$\leq \frac{(\nabla_i f(x_t))^2}{2(\lambda+\sqrt{w_{t,i}})} + \frac{\sigma_i^2}{c_i}\left(\mathbb{E}_{t-1}\left[\frac{g_{t,i}^2}{\lambda^2+v_{t,i}}\right]\right)^{\frac{2}{\bar p}} + c_i\mathbb{E}_{t-1}\left[\frac{g_{t,i}^2}{\lambda^2+v_{t,i}}\right], \tag{16}$$

where the last step is by $\lambda+\sqrt{w_{t,i}}\geq c_i$, $\frac{g_{t,i}^2}{\lambda^2+v_{t,i}}\leq 1$, and $\frac{\bar p}{2}\geq 1$.

We combine (12) and (16) to have

$$\mathbb{E}_{t-1}\left[f(x_{t+1})\right] \leq f(x_t) - \frac{\gamma}{2}\left\|\frac{(\nabla f(x_t))^2}{\lambda+\sqrt{w_t}}\right\|_1 + \gamma\sum_{i=1}^d \frac{\sigma_i^2}{c_i}\left(\mathbb{E}_{t-1}\left[\frac{g_{t,i}^2}{\lambda^2+v_{t,i}}\right]\right)^{\frac{2}{\bar p}}$$

$$+ \gamma\sum_{i=1}^d \left(c_i + \frac{\gamma L_i}{2}\right)\mathbb{E}_{t-1}\left[\frac{g_{t,i}^2}{\lambda^2+v_{t,i}}\right].$$

Taking expectations on both sides and rearranging terms, we know

$$\frac{\gamma}{2}\mathbb{E}\left[\left\|\frac{(\nabla f(x_t))^2}{\lambda+\sqrt{w_t}}\right\|_1\right] \leq \mathbb{E}\left[f(x_t)-f(x_{t+1})\right] + \gamma\sum_{i=1}^d \frac{\sigma_i^2}{c_i}\mathbb{E}\left[\left(\mathbb{E}_{t-1}\left[\frac{g_{t,i}^2}{\lambda^2+v_{t,i}}\right]\right)^{\frac{2}{\bar p}}\right]$$

$$+ \gamma\sum_{i=1}^d \left(c_i + \frac{\gamma L_i}{2}\right)\mathbb{E}\left[\frac{g_{t,i}^2}{\lambda^2+v_{t,i}}\right].$$

Finally, noticing that $\frac{\bar p}{2}\geq 1$, we hence can invoke Hölder's inequality again to have, for any $i\in[d]$,

$$\mathbb{E}\left[\left(\mathbb{E}_{t-1}\left[\frac{g_{t,i}^2}{\lambda^2+v_{t,i}}\right]\right)^{\frac{2}{\bar p}}\right] \leq \left(\mathbb{E}\left[\mathbb{E}_{t-1}\left[\frac{g_{t,i}^2}{\lambda^2+v_{t,i}}\right]\right]\right)^{\frac{2}{\bar p}} = \left(\mathbb{E}\left[\frac{g_{t,i}^2}{\lambda^2+v_{t,i}}\right]\right)^{\frac{2}{\bar p}},$$

which leads us to the desired result. $\qquad\square$

Lemma A.3 can be viewed as a generalization of the existing inequality in the literature (see, e.g., Ward et al. (2019)) from $q=1$ to any $q\in[0,1]$.

**Lemma A.3.** *For any $T\in\mathbb{N}$, $i\in[d]$, and $q\in[0,1]$, AdaGrad (Algorithm 1) guarantees*

$$\sum_{t=1}^T \left(\mathbb{E}\left[\frac{g_{t,i}^2}{\lambda^2+v_{t,i}}\right]\right)^q \leq T^{1-q}\left(\mathbb{E}\left[\ln\left(1+\frac{v_{T,i}}{\lambda^2}\right)\right]\right)^q.$$

*Proof.* By the concavity of $x^q$ (since $q\in[0,1]$), we have

$$\frac{1}{T}\sum_{t=1}^T \left(\mathbb{E}\left[\frac{g_{t,i}^2}{\lambda^2+v_{t,i}}\right]\right)^q \leq \left(\frac{1}{T}\sum_{t=1}^T \mathbb{E}\left[\frac{g_{t,i}^2}{\lambda^2+v_{t,i}}\right]\right)^q,$$

which implies that

$$
\sum_{t=1}^{T} \left( \mathbb{E}\left[ \frac{\boldsymbol{g}_{t,i}^2}{\lambda^2 + \boldsymbol{v}_{t,i}} \right] \right)^q \leq T^{1-q} \left( \mathbb{E}\left[ \sum_{t=1}^{T} \frac{\boldsymbol{g}_{t,i}^2}{\lambda^2 + \boldsymbol{v}_{t,i}} \right] \right)^q = T^{1-q} \left( \mathbb{E}\left[ \sum_{t=1}^{T} 1 - \frac{\lambda^2 + \boldsymbol{v}_{t-1,i}}{\lambda^2 + \boldsymbol{v}_{t,i}} \right] \right)^q
$$

$$
\overset{(a)}{\leq} T^{1-q} \left( \mathbb{E}\left[ \sum_{t=1}^{T} \ln\left( \frac{\lambda^2 + \boldsymbol{v}_{t,i}}{\lambda^2 + \boldsymbol{v}_{t-1,i}} \right) \right] \right)^q = T^{1-q} \left( \mathbb{E}\left[ \ln\left( 1 + \frac{\boldsymbol{v}_{T,i}}{\lambda^2} \right) \right] \right)^q,
$$

where $(a)$ is due to $1 - x^{-1} \leq \ln x, \forall x > 0$. $\qquad\square$

The next Lemma A.4 is the coordinate-wise version of Lemma 4.7.

**Lemma A.4.** *Under Assumption 2.4, for any $t \in \mathbb{N}$ and $i \in [d]$,* AdaGrad *(Algorithm 1) guarantees*

$$
\mathbb{E}\left[ \sqrt{\boldsymbol{v}_{t,i}} \right] \leq \sqrt{2}\boldsymbol{\sigma}_i t^{\frac{1}{p}} + \mathbb{E}\left[ \sqrt{2\boldsymbol{u}_{t,i}} \right],
$$

*where $\boldsymbol{u}_{t,i} \triangleq \sum_{s=1}^{t}(\nabla_i f(\boldsymbol{x}_s))^2, \forall t \in \mathbb{N}$.*

*Proof.* By the definition of $\boldsymbol{v}_{t,i}$, we have

$$
\sqrt{\boldsymbol{v}_{t,i}} = \sqrt{\sum_{s=1}^{t} \boldsymbol{g}_{s,i}^2} = \sqrt{\sum_{s=1}^{t}(\boldsymbol{\xi}_{s,i} + \nabla_i f(\boldsymbol{x}_s))^2} \leq \sqrt{2\sum_{s=1}^{t} \boldsymbol{\xi}_{s,i}^2 + 2\sum_{s=1}^{t}(\nabla_i f(\boldsymbol{x}_s))^2}
$$

$$
\leq \sqrt{2\sum_{s=1}^{t} \boldsymbol{\xi}_{s,i}^2} + \sqrt{2\boldsymbol{u}_{t,i}} \leq \sqrt{2}\left( \sum_{s=1}^{t} |\boldsymbol{\xi}_{s,i}|^p \right)^{\frac{1}{p}} + \sqrt{2\boldsymbol{u}_{t,i}},
$$

where the last step is due to $\|\cdot\|_2 \leq \|\cdot\|_p$ when $p \in [1,2]$. By Hölder's inequality, we conclude

$$
\mathbb{E}\left[ \sqrt{\boldsymbol{v}_{t,i}} \right] \leq \sqrt{2}\left( \mathbb{E}\left[ \sum_{s=1}^{t} |\boldsymbol{\xi}_{s,i}|^p \right] \right)^{\frac{1}{p}} + \mathbb{E}\left[ \sqrt{2\boldsymbol{u}_{t,i}} \right] \overset{\text{Assumption } 2.4}{\leq} \sqrt{2}\boldsymbol{\sigma}_i t^{\frac{1}{p}} + \mathbb{E}\left[ \sqrt{2\boldsymbol{u}_{t,i}} \right].
$$

$\qquad\square$

Finally, we prove Lemma A.5, which is also inspired by Ward et al. (2019).

**Lemma A.5.** *Under Assumptions 2.2 and 2.4, for any $i \in [d]$,* AdaGrad *(Algorithm 1) guarantees*

$$
\mathbb{E}\left[ \ln\left( 1 + \frac{\boldsymbol{v}_{T,i}}{\lambda^2} \right) \right] \leq 2\ln\left( 1 + \frac{\sqrt{2}\boldsymbol{\sigma}_i T^{\frac{1}{p}} + \mathbb{E}\left[ \sqrt{2\boldsymbol{u}_{T,i}} \right]}{\lambda} \right) \leq 2\ln K_T,
$$

*where $K_T \triangleq 1 + \frac{\sqrt{2}\|\boldsymbol{\sigma}\|_\infty T^{\frac{1}{p}} + 2\|\nabla f(\boldsymbol{x}_1)\|_\infty T^{\frac{1}{2}} + 2\gamma\sqrt{\|\boldsymbol{L}\|_1 \|\boldsymbol{L}\|_\infty} T^{\frac{3}{2}}}{\lambda}$.*

*Proof.* Note that

$$
\mathbb{E}\left[ \ln\left( 1 + \frac{\boldsymbol{v}_{T,i}}{\lambda^2} \right) \right] = 2\mathbb{E}\left[ \ln\left( \sqrt{1 + \frac{\boldsymbol{v}_{T,i}}{\lambda^2}} \right) \right] \leq 2\mathbb{E}\left[ \ln\left( 1 + \frac{\sqrt{\boldsymbol{v}_{T,i}}}{\lambda} \right) \right] \leq 2\ln\left( 1 + \frac{\mathbb{E}\left[ \sqrt{\boldsymbol{v}_{T,i}} \right]}{\lambda} \right),
$$

where the last step is due to the concavity of $\ln x$. Next, we invoke Lemma A.4 to obtain

$$
\mathbb{E}\left[ \ln\left( 1 + \frac{\boldsymbol{v}_{T,i}}{\lambda^2} \right) \right] \leq 2\ln\left( 1 + \frac{\sqrt{2}\boldsymbol{\sigma}_i T^{\frac{1}{p}} + \mathbb{E}\left[ \sqrt{2\boldsymbol{u}_{T,i}} \right]}{\lambda} \right). \tag{17}
$$

Moreover, under Assumption 2.2, we have almost surely, for any $t \in [T]$,

$$|\nabla_i f(\boldsymbol{x}_t) - \nabla_i f(\boldsymbol{x}_1)| \leq \sqrt{\boldsymbol{L}_i} \|\nabla f(\boldsymbol{x}_t) - \nabla f(\boldsymbol{x}_1)\|_{1/\boldsymbol{L}} \leq \sqrt{\boldsymbol{L}_i} \|\boldsymbol{x}_t - \boldsymbol{x}_1\|_{\boldsymbol{L}} \leq \sqrt{\boldsymbol{L}_i} \sum_{s=1}^{t-1} \|\boldsymbol{x}_{s+1} - \boldsymbol{x}_s\|_{\boldsymbol{L}}$$

$$= \sqrt{\boldsymbol{L}_i} \sum_{s=1}^{t-1} \left\| \frac{\gamma}{\lambda + \sqrt{\boldsymbol{v}_s}} \boldsymbol{g}_s \right\|_{\boldsymbol{L}} = \gamma \sqrt{\boldsymbol{L}_i} \sum_{s=1}^{t-1} \sqrt{\sum_{j=1}^{d} \frac{\boldsymbol{L}_j \boldsymbol{g}_{s,j}^2}{(\lambda + \sqrt{\boldsymbol{v}_{s,j}})^2}}$$

$$\leq \gamma \sqrt{\boldsymbol{L}_i} \sum_{s=1}^{t-1} \sqrt{\sum_{j=1}^{d} \boldsymbol{L}_j} = \gamma \sqrt{\boldsymbol{L}_i \|\boldsymbol{L}\|_1} (t-1)$$

$$\Rightarrow |\nabla_i f(\boldsymbol{x}_t)| \leq |\nabla_i f(\boldsymbol{x}_1)| + \gamma \sqrt{\boldsymbol{L}_i \|\boldsymbol{L}\|_1} (t-1).$$

Hence, there is almost surely

$$\sqrt{2\boldsymbol{u}_{T,i}} = \sqrt{2 \sum_{t=1}^{T} (\nabla_i f(\boldsymbol{x}_t))^2} \leq \sqrt{2 \sum_{t=1}^{T} \left( |\nabla_i f(\boldsymbol{x}_1)| + \gamma \sqrt{\boldsymbol{L}_i \|\boldsymbol{L}\|_1} (t-1) \right)^2}$$

$$\leq 2 \sqrt{\sum_{t=1}^{T} |\nabla_i f(\boldsymbol{x}_1)|^2 + \sum_{t=1}^{T} \gamma^2 \boldsymbol{L}_i \|\boldsymbol{L}\|_1 (t-1)^2}$$

$$\leq 2 |\nabla_i f(\boldsymbol{x}_1)| T^{\frac{1}{2}} + 2\gamma \sqrt{\boldsymbol{L}_i \|\boldsymbol{L}\|_1} T^{\frac{3}{2}}. \tag{18}$$

Finally, we plug (18) back into (17) to have

$$2 \ln \left( 1 + \frac{\sqrt{2} \boldsymbol{\sigma}_i T^{\frac{1}{p}} + \mathbb{E}\left[ \sqrt{2\boldsymbol{u}_{T,i}} \right]}{\lambda} \right)$$

$$\leq 2 \ln \left( 1 + \frac{\sqrt{2} \boldsymbol{\sigma}_i T^{\frac{1}{p}} + 2 |\nabla_i f(\boldsymbol{x}_1)| T^{\frac{1}{2}} + 2\gamma \sqrt{\boldsymbol{L}_i \|\boldsymbol{L}\|_1} T^{\frac{3}{2}}}{\lambda} \right)$$

$$\leq 2 \ln \left( 1 + \frac{\sqrt{2} \|\boldsymbol{\sigma}\|_\infty T^{\frac{1}{p}} + 2 \|\nabla f(\boldsymbol{x}_1)\|_\infty T^{\frac{1}{2}} + 2\gamma \sqrt{\|\boldsymbol{L}\|_1 \|\boldsymbol{L}\|_\infty} T^{\frac{3}{2}}}{\lambda} \right) = 2 \ln K_T.$$

$\square$

# B. Algorithm-Dependent Lower Bound for `AdaGrad`

This section provides the full statement of Theorem 3.3 and its proof.

## B.1. Full Theorem and Its Proof

**Theorem B.1** (Full statement of Theorem 3.3). *Let $d = 1$, for any given $\Delta > 0$, $L > 0$, $p \in (1, 2]$, $\sigma \geq 0$, $0 < \epsilon \leq \sqrt{2\Delta L}$, $x_1 \in \mathbb{R}$, $\gamma > 0$, $\lambda \geq 0$ satisfying $\lambda = 0$ when $\sigma = 0$, there exists a function $f : \mathbb{R} \to \mathbb{R}$ associated with a function $g : \mathbb{R} \times \{0, 1\} \to \mathbb{R}$ and a Bernoulli distribution $\mathbb{P}$ on $\{0, 1\}$ satisfying*

1. *$f(x_1) - \inf_{x \in \mathbb{R}} f(x) \leq \Delta$ and $f$ is $L$-smooth;*

2. *$\mathbb{E}_{r \sim \mathbb{P}} [g(x; r)] = f'(x)$ and $\mathbb{E}_{r \sim \mathbb{P}} \left[ |g(x; r) - f'(x)|^p \right] \leq \sigma^p$.*

*Moreover, if using* `AdaGrad` *(Algorithm 1), with initial point $x_1$, learning rate $\gamma$, and hyperparameter $\lambda$, to optimize $f$ by interacting with $g$ (i.e., $g_t = g(x_t; r_t)$ where $r_t \sim \mathbb{P}$ is independent of the history), one must use at least*

$$\Omega \left( \frac{\lambda \Delta / \gamma + \Delta^2/\gamma^2 + \gamma^2 L^2 \ln^2 \left( \frac{\gamma L (1 + \sigma^{\frac{p}{p-1}} \epsilon^{-\frac{p}{p-1}})}{\lambda + \epsilon(1 + \sigma^{\frac{p}{p-1}} \epsilon^{-\frac{p}{p-1}})} \right)}{\epsilon^2} + \frac{\left( \Delta^2/\gamma^2 + \gamma^2 L^2 \ln^2 \left( \frac{\gamma L (1 + \sigma^{\frac{p}{p-1}} \epsilon^{-\frac{p}{p-1}})}{\lambda + \epsilon(1 + \sigma^{\frac{p}{p-1}} \epsilon^{-\frac{p}{p-1}})} \right) \right) \sigma^{\frac{p}{p-1}}}{\epsilon^{\frac{3p-2}{p-1}}} \right)$$

*iterations to make $\mathbb{E} \left[ \frac{1}{T} \sum_{t=1}^{T} |f'(x_t)| \right] < \frac{\epsilon}{2}$ for small enough $\epsilon$ (see (28) for the precise condition on $\epsilon$).*

*Remark* B.2. The technical condition $\lambda = 0$ when $\sigma = 0$ is imposed to ensure that the second and third inequalities in (28) hold in the deterministic case. In fact, it suffices to require $\lambda \leq \mathcal{O}(\epsilon)$ when $\sigma = 0$, but we set $\lambda = 0$ for simplicity.

*Proof.* In the proof, we write

$$q \triangleq \frac{1}{\left[ 1 + \frac{p-1}{4} \left( \frac{2\sigma}{\epsilon} \right)^p \right]^{\frac{1}{p-1}}} \in [0, 1]. \tag{19}$$

Moreover, let

$$\delta_t \triangleq \frac{\gamma}{\frac{\lambda q}{\epsilon} + \sqrt{t}}, \tag{20}$$

and

$$T_\star \triangleq \inf \left\{ T \in \mathbb{N} : \Delta - \epsilon \sum_{t=1}^{T} \delta_t + \frac{L}{4} \sum_{t=1}^{T} \delta_t^2 < \frac{\epsilon^2}{2L} \right\}. \tag{21}$$

Now, let us consider the function $f$ constructed in Lemma B.3 and the following function $g : \mathbb{R} \times \{0, 1\} \to \mathbb{R}$,

$$g(x; r) = \begin{cases} f'(x) & x \notin \{y_1, \ldots, y_{T_\star}\} \\ \frac{r}{q} f'(x) & x \in \{y_1, \ldots, y_{T_\star}\} \end{cases}, \tag{22}$$

where we recall that $y_t = x_1 + \sum_{s=1}^{t-1} \delta_s, \forall t \in [T_\star]$ is introduced in Lemma B.3 satisfying

$$f'(y_t) = -\epsilon, \forall t \in [T_\star]. \tag{23}$$

According to Lemma B.3, we know

$$f(x_1) - \inf_{x \in \mathbb{R}} f(x) \leq \Delta \quad \text{and} \quad f \text{ is } L\text{-smooth.}$$

Next, let $\mathbb{P}$ be the Bernoulli distribution with the parameter $q$ given in (19), i.e.,

$$\mathbb{P}[r = 0] = 1 - q \quad \text{and} \quad \mathbb{P}[r = 1] = q. \tag{24}$$

One can find that $\mathbb{E}_{r \sim \mathbb{P}}[g(x; r)] \overset{(22),(24)}{=} f'(x)$ and $\mathbb{E}_{r \sim \mathbb{P}} \left[ |g(x; r) - f'(x)|^p \right] \overset{(22)}{=} 0 \leq \sigma^p$ if $x \notin \{y_1, \ldots, y_{T_\star}\}$. If $x \in \{y_1, \ldots, y_{T_\star}\}$, we know

$$\mathbb{E}_{r \sim \mathbb{P}} \left[ |g(x; r) - f'(x)|^p \right] \overset{(22),(24)}{=} |f'(x)|^p (1 - q) + \left| \frac{f'(x)}{q} - f'(x) \right|^p q \overset{(23)}{=} (1 - q) \frac{q^{p-1} + (1 - q)^{p-1}}{q^{p-1}} \epsilon^p$$

$$\overset{(a)}{\leq} (1 - q) \frac{2^{2-p} \epsilon^p}{q^{p-1}} \overset{(b)}{\leq} \frac{(1 - q^{p-1}) 2^{2-p} \epsilon^p}{(p-1) q^{p-1}} \overset{(19)}{=} \sigma^p,$$

where $(a)$ is by $\frac{q^{p-1} + (1-q)^{p-1}}{2} \leq \frac{1}{2^{p-1}}$ due to the concavity of $x^{p-1}$ (since $p - 1 \in (0, 1]$) and $(b)$ holds by $1 - q \leq \frac{1 - q^{p-1}}{p-1}, \forall q \in [0, 1], p \in (1, 2]$. Therefore, we know

$$\mathbb{E}_{r \sim \mathbb{P}}[g(x; r)] = f'(x) \quad \text{and} \quad \mathbb{E}_{r \sim \mathbb{P}} \left[ |g(x; r) - f'(x)|^p \right] \leq \sigma^p.$$

Suppose one runs `AdaGrad` to optimize $f$ by interacting with $g$. Let us define

$$R_t \triangleq \sum_{s=1}^{t} r_t, \forall t \in \mathbb{N} \quad \text{and} \quad E_T \triangleq \{R_T \leq T_\star - 1\}, \forall T \in \mathbb{N}. \tag{25}$$

Given $T \in \mathbb{N}$, for any sample path in $E_T$, we use induction to show

$$\{x_1, \ldots, x_t\} \subseteq \{y_1, \ldots, y_{T_\star}\}, \forall t \in [T]. \tag{26}$$

For $t = 1$, (26) is true since $y_1 = x_1$. Suppose (26) holds for some $t \leq T - 1$, then we know

$$g_s = g(x_s; r_s) \stackrel{(22),(23),(26)}{=} -\frac{r_s}{q}\epsilon, \forall s \in [t],$$

which implies that

$$v_s = \sum_{\ell=1}^{s} g_\ell^2 = \frac{\epsilon^2}{q^2}\sum_{\ell=1}^{s} r_\ell^2 = \frac{\epsilon^2}{q^2}\sum_{\ell=1}^{s} r_\ell \stackrel{(25)}{=} \frac{\epsilon^2}{q^2}R_s, \forall s \in [t].$$

Therefore, by the update rule of `AdaGrad`,

$$x_{t+1} = x_1 - \sum_{s=1}^{t} \frac{\gamma}{\lambda + \sqrt{v_s}}g_s = x_1 + \sum_{s=1}^{t} \frac{\gamma r_s}{\frac{\lambda q}{\epsilon} + \sqrt{R_s}} \stackrel{(20)}{=} x_1 + \sum_{s=1}^{t} \delta_{R_s}r_s = x_1 + \sum_{s=1}^{R_t} \delta_s \in \{y_1, \ldots, y_{T_\star}\},$$

where the last step is due to $R_t \leq R_T \leq T_\star - 1$ and $y_t = x_1 + \sum_{s=1}^{t-1} \delta_s, \forall t \in [T_\star]$. Thus, the induction is complete. If $T \leq \frac{T_\star - 1}{2q}$, by Markov's inequality, we have

$$\mathbb{P}[R_T > 2qT] \leq \frac{\mathbb{E}[R_T]}{2qT} = \frac{1}{2} \Rightarrow \mathbb{P}[E_T] \geq \mathbb{P}[R_T \leq 2qT] \geq \frac{1}{2},$$

which implies that

$$\mathbb{E}\left[\frac{1}{T}\sum_{t=1}^{T}|f'(x_t)|\right] \geq \mathbb{E}\left[\frac{1}{T}\sum_{t=1}^{T}|f'(x_t)| \mid E_T\right]\mathbb{P}[E_T] \geq \frac{\epsilon}{2},$$

where the last step is due to $|f'(x_t)| = \epsilon, \forall t \in [T]$ since $\{x_1, \ldots, x_T\} \subseteq \{y_1, \ldots, y_{T_\star}\}$ (see (26)) and $|f'(y_t)| = \epsilon, \forall t \in [T_\star]$ (see (23)). Therefore, to make $\mathbb{E}\left[\frac{1}{T}\sum_{t=1}^{T}|f'(x_t)|\right] < \frac{\epsilon}{2}$, one must have

$$T > \frac{T_\star - 1}{2q}. \tag{27}$$

Finally, let us assume $\epsilon$ is small enough to satisfy[4]

$$\epsilon \leq \sqrt{\Delta L}, \quad \ln\left(1 + \frac{1}{\left(\frac{\lambda q}{\epsilon}\right)^2 + 1}\right) \geq \frac{16\epsilon}{\gamma L}, \quad \frac{16\epsilon}{\gamma L}\sqrt{\left(\frac{\lambda q}{\epsilon}\right)^2 + 1} \leq \frac{\ln c}{\sqrt{2c}}, \tag{28}$$

---

[4]The first condition is trivial to satisfy. To make the second one hold, since $\ln(1+\frac{1}{x}) \geq \frac{1}{x+1}$, it suffices to ensure that $\frac{16\epsilon}{\gamma L}\left(\left(\frac{\lambda q}{\epsilon}\right)^2 + 2\right) \leq 1$, which is possible when $\epsilon$ is small enough due to the definition of $q$ (see (19)). The third one is also true when $\epsilon$ is small enough by the definition of $q$.

where $c \in [3.92, 3.93]$ is the unique positive solution to $2c = (1+c)\ln(1+c)$. Note that

$$T_\star \overset{(20),(21)}{=} \inf\left\{T \in \mathbb{N} : \Delta - \epsilon\gamma\sum_{t=1}^{T}\frac{1}{\frac{\lambda q}{\epsilon} + \sqrt{t}} + \frac{\gamma^2 L}{4}\sum_{t=1}^{T}\frac{1}{\left(\frac{\lambda q}{\epsilon} + \sqrt{t}\right)^2} < \frac{\epsilon^2}{2L}\right\}$$

$$\overset{(28)}{\geq} \inf\left\{T \in \mathbb{N} : \frac{\Delta}{2} - \epsilon\gamma\sum_{t=1}^{T}\frac{1}{\frac{\lambda q}{\epsilon} + \sqrt{t}} + \frac{\gamma^2 L}{4}\sum_{t=1}^{T}\frac{1}{\left(\frac{\lambda q}{\epsilon} + \sqrt{t}\right)^2} < 0\right\}$$

$$= \inf\left\{T \in \mathbb{N} : \frac{\Delta}{2\gamma\epsilon} + \frac{\gamma L}{4\epsilon}\sum_{t=1}^{T}\frac{1}{\left(\frac{\lambda q}{\epsilon} + \sqrt{t}\right)^2} < \sum_{t=1}^{T}\frac{1}{\frac{\lambda q}{\epsilon} + \sqrt{t}}\right\}.$$

Moreover, we observe that

$$\sum_{t=1}^{T}\frac{1}{\frac{\lambda q}{\epsilon} + \sqrt{t}} \leq \int_0^T \frac{\mathrm{d}t}{\frac{\lambda q}{\epsilon} + \sqrt{t}} \leq \int_0^T \frac{\mathrm{d}t}{\sqrt{\left(\frac{\lambda q}{\epsilon}\right)^2 + t}} = 2\sqrt{\left(\frac{\lambda q}{\epsilon}\right)^2 + T} - \frac{2\lambda q}{\epsilon},$$

$$\sum_{t=1}^{T}\frac{1}{\left(\frac{\lambda q}{\epsilon} + \sqrt{t}\right)^2} \geq \int_1^{T+1}\frac{\mathrm{d}t}{\left(\frac{\lambda q}{\epsilon} + \sqrt{t}\right)^2} \geq \int_1^{T+1}\frac{\mathrm{d}t}{2\left(\left(\frac{\lambda q}{\epsilon}\right)^2 + t\right)} = \frac{\ln\left(1 + \frac{T}{\left(\frac{\lambda q}{\epsilon}\right)^2 + 1}\right)}{2},$$

which together imply that

$$T_\star \geq \inf\left\{T \in \mathbb{N} : \frac{\Delta}{2\gamma\epsilon} + \frac{\gamma L}{8\epsilon}\ln\left(1 + \frac{T}{\left(\frac{\lambda q}{\epsilon}\right)^2 + 1}\right) < 2\sqrt{\left(\frac{\lambda q}{\epsilon}\right)^2 + T} - \frac{2\lambda q}{\epsilon}\right\}$$

$$\geq \frac{\inf\left\{T \in \mathbb{N} : \frac{\Delta}{4\gamma\epsilon} < \sqrt{\left(\frac{\lambda q}{\epsilon}\right)^2 + T} - \frac{\lambda q}{\epsilon}\right\} + \inf\left\{T \in \mathbb{N} : \frac{\ln\left(1 + \frac{T}{\left(\frac{\lambda q}{\epsilon}\right)^2 + 1}\right)}{\sqrt{T}} < \frac{16\epsilon}{\gamma L}\right\}}{2}$$

$$\geq \frac{\left\lceil\frac{\lambda\Delta q}{2\gamma\epsilon^2} + \frac{\Delta^2}{16\gamma^2\epsilon^2}\right\rceil + \left\lceil\frac{\gamma^2 L^2}{64\epsilon^2}\ln^2\left(\frac{\gamma L}{8\epsilon\sqrt{\left(\frac{\lambda q}{\epsilon}\right)^2 + 1}}\right)\right\rceil}{2}, \tag{29}$$

where the last step is due to (28) and Lemma B.4. Therefore, we obtain

$$T \overset{(27),(29)}{\geq} \Omega\left(\frac{\lambda\Delta}{\gamma\epsilon^2} + \frac{\Delta^2/\gamma^2 + \gamma^2 L^2\ln^2\left(\frac{\gamma L}{\lambda q + \epsilon}\right)}{\epsilon^2 q}\right)$$

$$\overset{(19)}{=} \Omega\left(\frac{\lambda\Delta/\gamma + \Delta^2/\gamma^2 + \gamma^2 L^2\ln^2\left(\frac{\gamma L(1+\sigma^{\frac{p}{p-1}}\epsilon^{-\frac{p}{p-1}})}{\lambda+\epsilon(1+\sigma^{\frac{p}{p-1}}\epsilon^{-\frac{p}{p-1}})}\right)}{\epsilon^2} + \frac{\left(\Delta^2/\gamma^2 + \gamma^2 L^2\ln^2\left(\frac{\gamma L(1+\sigma^{\frac{p}{p-1}}\epsilon^{-\frac{p}{p-1}})}{\lambda+\epsilon(1+\sigma^{\frac{p}{p-1}}\epsilon^{-\frac{p}{p-1}})}\right)\right)\sigma^{\frac{p}{p-1}}}{\epsilon^{\frac{3p-2}{p-1}}}\right).$$

$\square$

## B.2. Helpful Lemmas

We provide three technical lemmas used in proving the algorithm-dependent lower bound for `AdaGrad`.

We first prove Lemma B.3, which is essentially Lemma 20 of Hübler et al. (2025) (see also Lemma 15 of Jiang et al. (2025)).

**Lemma B.3.** *Given $\Delta > 0$, $L > 0$, $0 < \epsilon \leq \sqrt{2\Delta L}$, $x_1 \in \mathbb{R}$, and a nonnegative sequence $\{\delta_t\}_{t=1}^\infty$, let*

$$T_\star \triangleq \inf\left\{T \in \mathbb{N} : \Delta - \epsilon\sum_{t=1}^T \delta_t + \frac{L}{4}\sum_{t=1}^T \delta_t^2 < \frac{\epsilon^2}{2L}\right\},$$

*and $y_t \triangleq x_1 + \sum_{s=1}^{t-1}\delta_s, \forall t \in [T_\star]$, then there exists a function $f : \mathbb{R} \to \mathbb{R}$ such that:*

$$f(x_1) - \inf_{x \in \mathbb{R}} f(x) \leq \Delta; \quad f \text{ is } L\text{-smooth}; \quad f'(y_t) = -\epsilon, \forall t \in [T_\star].$$

*Proof.* For any $\delta \geq 0$, let

$$g_\delta(x) \triangleq \begin{cases} -\epsilon + Lx & x \in [0, \delta/2] \\ -\epsilon + L\delta - Lx & x \in (\delta/2, \delta] \end{cases}.$$

Now, we introduce

$$f'(x) \triangleq \begin{cases} -\epsilon & x < y_1 \\ g_{\delta_t}(x - y_t) & x \in [y_t, y_{t+1}), t \in [T_\star - 1] \\ -\epsilon + L(x - y_{T_\star}) & x \geq y_{T_\star} \end{cases},$$

and

$$f(x) \triangleq \Delta + \int_{y_1}^x f'(z)\mathrm{d}z.$$

Note that $f$ is $L$-smooth and satisfies $f'(y_t) = -\epsilon, \forall t \in [T_\star]$ by its definition. Thus, we only need to verify $f(x_1) - \inf_{x \in \mathbb{R}} f(x) \leq \Delta$. Note that $f(x_1) = \Delta + \int_{y_1}^{x_1} f'(z)\mathrm{d}z = \Delta$ due to $y_1 = x_1$, it remains to show $\inf_{x \in \mathbb{R}} f(x) \geq 0$.

First, we can find that

$$f(x) = \Delta + \epsilon(y_1 - x), \forall x < y_1 \Rightarrow \inf_{x < y_1} f(x) = f(y_1).$$

Next, given $t \in [T_\star - 1]$, we can find that

$$\inf_{x \in [y_t, y_{t+1})} f(x) = \begin{cases} \min\left\{f(y_t) - \frac{\epsilon^2}{2L}, f(y_{t+1})\right\} & \frac{\delta_t}{2} \geq \frac{\epsilon}{L} \\ f(y_{t+1}) & \frac{\delta_t}{2} < \frac{\epsilon}{L} \end{cases} \geq \min\left\{f(y_t) - \frac{\epsilon^2}{2L}, f(y_{t+1})\right\}.$$

Finally, we know

$$\inf_{x \geq y_{T_\star}} f(x) = f(y_{T_\star} + \epsilon/L) = f(y_{T_\star}) - \frac{\epsilon^2}{2L}.$$

The above three results together imply that

$$\inf_{x \in \mathbb{R}} f(x) \geq \min_{t \in [T_\star]} f(y_t) - \frac{\epsilon^2}{2L}.$$

Now, we compute

$$f(y_t) = \Delta + \sum_{s=1}^{t-1}\int_{y_s}^{y_{s+1}} g_{\delta_s}(z - y_s)\mathrm{d}z = \Delta - \epsilon\sum_{s=1}^{t-1}\delta_s + \frac{L}{4}\sum_{s=1}^{t-1}\delta_s^2, \forall t \in [T_\star],$$

which implies that, by the definition of $T_\star$,

$$\min_{t \in [T_\star]} f(y_t) - \frac{\epsilon^2}{2L} = \min_{t \in [T_\star]}\Delta - \epsilon\sum_{s=1}^{t-1}\delta_s + \frac{L}{4}\sum_{s=1}^{t-1}\delta_s^2 - \frac{\epsilon^2}{2L} \geq 0.$$

$\square$

Next, Lemma B.4 provides a lower bound for an important quantity used in the proof of Theorem B.1.

**Lemma B.4.** *Let $c \in [3.92, 3.93]$ denote the unique positive solution to $2c = (1 + c)\ln(1 + c)$, given $A > 0$ and $B \geq 1$ satisfying $\ln\left(1 + \frac{1}{B}\right) \geq A$ and $A\sqrt{B} \leq \frac{\ln c}{\sqrt{2c}}$, we have*

$$\inf\left\{T \in \mathbb{N} : \frac{\ln\left(1 + \frac{T}{B}\right)}{\sqrt{T}} < A\right\} \geq \left\lceil \frac{4}{A^2} \ln^2\left(\frac{2}{A\sqrt{B}}\right)\right\rceil.$$

*Proof.* Let $h(x) \triangleq \frac{\ln\left(1 + \frac{x}{B}\right)}{\sqrt{x}}$ for $x > 0$. We have

$$h'(x) = \frac{B\left[\frac{2x}{B} - \left(1 + \frac{x}{B}\right)\ln\left(1 + \frac{x}{B}\right)\right]}{2x^{\frac{3}{2}}(B + x)}.$$

We can find $h'(x) \geq 0 \Leftrightarrow \frac{2x}{B} - \left(1 + \frac{x}{B}\right)\ln\left(1 + \frac{x}{B}\right) \geq 0 \Leftrightarrow x \in (0, cB]$. Therefore, $h(T) \geq h(1) = \ln\left(1 + \frac{1}{B}\right) \geq A$ when $T \in [\lfloor cB \rfloor]$, implying that

$$\inf\left\{T \in \mathbb{N} : \frac{\ln\left(1 + \frac{T}{B}\right)}{\sqrt{T}} < A\right\} = \inf\left\{\lfloor cB \rfloor + 1 \leq T \in \mathbb{N} : \frac{\ln\left(1 + \frac{T}{B}\right)}{\sqrt{T}} < A\right\}$$

$$\geq \inf\left\{\lfloor cB \rfloor + 1 \leq T \in \mathbb{N} : \frac{\ln\left(\frac{T}{B}\right)}{\sqrt{T}} < A\right\}$$

$$= \inf\left\{\lfloor cB \rfloor + 1 \leq T \in \mathbb{N} : -\frac{A\sqrt{T}}{2}\exp\left(-\frac{A\sqrt{T}}{2}\right) > -\frac{A\sqrt{B}}{2}\right\}.$$

Now, we redefine $h(x) \triangleq -x\exp(-x)$ for $x \geq 0$. Note that $h'(x) = \exp(-x)(x - 1) \Rightarrow \min_{x \geq 0} h(x) = h(1) = -\frac{1}{e}$. Next, we observe that

$$h\left(\frac{A\sqrt{\lfloor cB \rfloor + 1}}{2}\right) \leq -\frac{A\sqrt{B}}{2} \Leftrightarrow \sqrt{\lfloor cB \rfloor + 1} \geq \sqrt{B}\exp\left(\frac{A\sqrt{\lfloor cB \rfloor + 1}}{2}\right),$$

which holds due to

$$\sqrt{\lfloor cB \rfloor + 1} \geq \sqrt{cB} \overset{A\sqrt{B} \leq \frac{\ln c}{\sqrt{2c}}}{\geq} \sqrt{B}\exp\left(\frac{A\sqrt{2cB}}{2}\right) \overset{cB \geq \lfloor cB \rfloor \geq 1}{\geq} \sqrt{B}\exp\left(\frac{A\sqrt{\lfloor cB \rfloor + 1}}{2}\right).$$

Hence,

$$\inf\left\{\lfloor cB \rfloor + 1 \leq T \in \mathbb{N} : -\frac{A\sqrt{T}}{2}\exp\left(-\frac{A\sqrt{T}}{2}\right) > -\frac{A\sqrt{B}}{2}\right\} \geq \lceil T_{\text{root}} \rceil,$$

where $T_{\text{root}} \in \mathbb{R}$ is the unique solution of $h\left(\frac{A\sqrt{T}}{2}\right) = -\frac{A\sqrt{B}}{2}$ that guarantees $\frac{A\sqrt{T_{\text{root}}}}{2} > 1$. More precisely, let $W_{-1}$ be the Lambert $W$ function, we have

$$\frac{A\sqrt{T_{\text{root}}}}{2} = -W_{-1}\left(-\frac{A\sqrt{B}}{2}\right) \Leftrightarrow T_{\text{root}} = \frac{4}{A^2}\left[-W_{-1}\left(-\frac{A\sqrt{B}}{2}\right)\right]^2.$$

Finally, we apply the standard inequality $-W_{-1}(-x) \geq \ln\frac{1}{x}$ to conclude. $\square$

Lastly, we prove Lemma B.5, which can help us further lower bound the algorithm-dependent lower bound.

**Lemma B.5.** *Given $A > 0$, we have $\inf_{\eta > 0} \frac{1}{\eta} + \eta\ln^2(A\eta) \geq \ln A$.*

*Proof.* We fix $\eta > 0$ and define $h_\eta(x) \triangleq \eta x^2 - (1 - 2\eta\ln\eta)x + \eta\ln^2\eta + \frac{1}{\eta}$. Note that $\eta > 0$ and the discriminant of $h_\eta$ is $-4\eta\ln\eta - 3 \leq 4/e - 3 < 0$, since $\min_{\eta > 0}\eta\ln\eta = -1/e$. Therefore, $h_\eta(x) \geq 0$ for all $x \in \mathbb{R}$. In particular, we have

$$\frac{1}{\eta} + \eta\ln^2(A\eta) - \ln A = h_\eta(\ln A) \geq 0,$$

which implies the desired result. $\square$

# C. Another Upper Bound for `AdaGrad-Norm`

In this section, we provide another upper bound for `AdaGrad-Norm`, given in Theorem C.1. Unlike Theorem 4.2, this bound does not require the objective function to be bounded. However, it is only in the order of $\tilde{\mathcal{O}}(1/T^{\frac{3p-4}{4p}})$ as a trade-off (same as Theorem A.1 for `AdaGrad`), which becomes vacuous when $p \in (1, ^4/_3]$.

## C.1. Theorem and Its Proof

**Theorem C.1.** *Under Assumptions 2.1, 2.2, 2.3, and 2.4, let $\Delta \triangleq f(\boldsymbol{x}_1) - f_\star$, then for any $\gamma > 0$ and $\lambda > 0$, `AdaGrad-Norm` (Algorithm 2) guarantees*

$$
\mathbb{E}\left[\frac{1}{T}\sum_{t=1}^{T}\|\nabla f(\boldsymbol{x}_t)\|_2\right] \leq \mathcal{O}\left(\frac{\lambda + \frac{\Delta}{\gamma} + \gamma\|\boldsymbol{L}\|_\infty \ln K_T}{\sqrt{T}} + \frac{\|\boldsymbol{\sigma}\|_p \ln^{\frac{1}{p}+\frac{1}{2}} K_T}{T^{\frac{p-1}{p}}}\right.
$$
$$
\left. + \frac{\sqrt{\left(\frac{\Delta}{\gamma} + \gamma\|\boldsymbol{L}\|_\infty \ln K_T\right)\|\boldsymbol{\sigma}\|_p}}{T^{\frac{p-1}{2p}}} + \frac{\|\boldsymbol{\sigma}\|_p \ln^{\frac{1}{2p}+\frac{1}{4}} K_T}{T^{\frac{3p-4}{4p}}}\right),
$$

*where $K_T = 1 + \frac{\sqrt{2}\|\boldsymbol{\sigma}\|_p T^{\frac{1}{p}} + 2\|\nabla f(\boldsymbol{x}_1)\|_2 T^{\frac{1}{2}} + 2\gamma\|\boldsymbol{L}\|_\infty T^{\frac{3}{2}}}{\lambda}$ is introduced in Lemma C.5.*

*Proof.* Equipped with Lemmas C.2 (choose $c \triangleq \|\boldsymbol{\sigma}\|_p T^{\frac{1}{2}-\frac{1}{p}}/D_T^{\frac{1}{2}-\frac{1}{p}}$ for $D_T \triangleq 2\ln(1 + \frac{\sqrt{2}\|\boldsymbol{\sigma}\|_p T^{\frac{1}{p}} + \mathbb{E}[\sqrt{2u_T}]}{\lambda})$ when invoking it), C.3, C.4, and C.5, the proof of Theorem C.1 follows essentially the same way as proving Theorem A.1, which is omitted here to save space. $\square$

## C.2. Helpful Lemmas

This subsection provides all necessary lemmas to prove Theorem C.1. The following four lemmas correspond to Lemmas A.2, A.3, A.4, and A.5 under the $\ell_2$ geometry. Their proofs do not involve new techniques, except that $w_t$ is now defined as follows

$$
w_t \triangleq v_{t-1} + \|\nabla f(\boldsymbol{x}_t)\|_2^2 + c^2 \in \mathcal{F}_{t-1}, \forall t \in \mathbb{N},
$$

where $c \geq 0$ can be an arbitrary constant and will be determined in the proof of Theorem C.1.

**Lemma C.2.** *Under Assumptions 2.2, 2.3, and 2.4, for any $c \geq 0$ and $t \in \mathbb{N}$, `AdaGrad-Norm` (Algorithm 2) guarantees*

$$
\frac{\gamma}{2}\mathbb{E}\left[\frac{\|\nabla f(\boldsymbol{x}_t)\|_2^2}{\lambda + \sqrt{w_t}}\right] \leq \mathbb{E}\left[f(\boldsymbol{x}_t) - f(\boldsymbol{x}_{t+1})\right] + \gamma\frac{\|\boldsymbol{\sigma}\|_p^2}{c}\left(\mathbb{E}\left[\frac{\|\boldsymbol{g}_t\|_2^2}{\lambda^2 + v_t}\right]\right)^{\frac{2}{p}} + \gamma\left(c + \frac{\gamma\|\boldsymbol{L}\|_\infty}{2}\right)\mathbb{E}\left[\frac{\|\boldsymbol{g}_t\|_2^2}{\lambda^2 + v_t}\right].
$$

*Proof.* We start with Assumption 2.2 and use the update rule of `AdaGrad-Norm` to obtain

$$
f(\boldsymbol{x}_{t+1}) \leq f(\boldsymbol{x}_t) + \langle\nabla f(\boldsymbol{x}_t), \boldsymbol{x}_{t+1} - \boldsymbol{x}_t\rangle + \frac{\|\boldsymbol{x}_{t+1} - \boldsymbol{x}_t\|_{\boldsymbol{L}}^2}{2}
$$
$$
= f(\boldsymbol{x}_t) - \gamma\left\langle\nabla f(\boldsymbol{x}_t), \frac{\boldsymbol{g}_t}{\lambda + \sqrt{v_t}}\right\rangle + \frac{\gamma^2\|\boldsymbol{g}_t\|_{\boldsymbol{L}}^2}{2(\lambda + \sqrt{v_t})^2}
$$
$$
\leq f(\boldsymbol{x}_t) - \gamma\left\langle\nabla f(\boldsymbol{x}_t), \frac{\boldsymbol{g}_t}{\lambda + \sqrt{v_t}}\right\rangle + \frac{\gamma^2\|\boldsymbol{L}\|_\infty\|\boldsymbol{g}_t\|_2^2}{2(\lambda^2 + v_t)}
$$
$$
= f(\boldsymbol{x}_t) - \gamma\left\langle\nabla f(\boldsymbol{x}_t), \frac{\boldsymbol{g}_t}{\lambda + \sqrt{w_t}}\right\rangle + \gamma\left\langle\nabla f(\boldsymbol{x}_t), \frac{\boldsymbol{g}_t}{\lambda + \sqrt{w_t}} - \frac{\boldsymbol{g}_t}{\lambda + \sqrt{v_t}}\right\rangle + \frac{\gamma^2\|\boldsymbol{L}\|_\infty\|\boldsymbol{g}_t\|_2^2}{2(\lambda^2 + v_t)}.
$$

Take conditional expectations on both sides of the above inequality and use

$$
\mathbb{E}_{t-1}\left[\left\langle\nabla f(\boldsymbol{x}_t), \frac{\boldsymbol{g}_t}{\lambda + \sqrt{w_t}}\right\rangle\right] \overset{\boldsymbol{x}_t, w_t \in \mathcal{F}_{t-1}}{=} \left\langle\nabla f(\boldsymbol{x}_t), \frac{\mathbb{E}_{t-1}[\boldsymbol{g}_t]}{\lambda + \sqrt{w_t}}\right\rangle \overset{\text{Assumption } 2.3}{=} \frac{\|\nabla f(\boldsymbol{x}_t)\|_2^2}{\lambda + \sqrt{w_t}}
$$

to obtain

$$\mathbb{E}_{t-1}\left[f(\boldsymbol{x}_{t+1})\right] \leq f(\boldsymbol{x}_t) - \gamma \frac{\|\nabla f(\boldsymbol{x}_t)\|_2^2}{\lambda + \sqrt{w_t}} + \frac{\gamma^2 \|\boldsymbol{L}\|_\infty}{2} \mathbb{E}_{t-1}\left[\frac{\|\boldsymbol{g}_t\|_2^2}{\lambda^2 + v_t}\right]$$
$$+ \gamma \mathbb{E}_{t-1}\left[\left\langle \nabla f(\boldsymbol{x}_t), \frac{\boldsymbol{g}_t}{\lambda + \sqrt{w_t}} - \frac{\boldsymbol{g}_t}{\lambda + \sqrt{v_t}} \right\rangle\right]. \tag{30}$$

Now, we can bound

$$\mathbb{E}_{t-1}\left[\left\langle \nabla f(\boldsymbol{x}_t), \frac{\boldsymbol{g}_t}{\lambda + \sqrt{w_t}} - \frac{\boldsymbol{g}_t}{\lambda + \sqrt{v_t}} \right\rangle\right]$$
$$\leq \mathbb{E}_{t-1}\left[\|\nabla f(\boldsymbol{x}_t)\|_2 \|\boldsymbol{g}_t\|_2 \left|\frac{1}{\lambda + \sqrt{w_t}} - \frac{1}{\lambda + \sqrt{v_t}}\right|\right]$$
$$\overset{(a)}{=} \frac{\|\nabla f(\boldsymbol{x}_t)\|_2}{\lambda + \sqrt{w_t}} \mathbb{E}_{t-1}\left[\frac{\|\boldsymbol{g}_t\|_2 \left|\|\boldsymbol{g}_t\|_2^2 - \|\nabla f(\boldsymbol{x}_t)\|_2^2 - c^2\right|}{(\lambda + \sqrt{v_t})(\sqrt{w_t} + \sqrt{v_t})}\right]$$
$$\leq \frac{\|\nabla f(\boldsymbol{x}_t)\|_2}{\lambda + \sqrt{w_t}} \left(\mathbb{E}_{t-1}\left[\frac{(\|\boldsymbol{g}_t\|_2 + \|\nabla f(\boldsymbol{x}_t)\|_2)\|\boldsymbol{\xi}_t\|_2 \|\boldsymbol{g}_t\|_2}{(\lambda + \sqrt{v_t})(\sqrt{w_t} + \sqrt{v_t})}\right] + \mathbb{E}_{t-1}\left[\frac{c^2 \|\boldsymbol{g}_t\|_2}{(\lambda + \sqrt{v_t})(\sqrt{w_t} + \sqrt{v_t})}\right]\right)$$
$$\overset{(b)}{\leq} \frac{\|\nabla f(\boldsymbol{x}_t)\|_2}{\lambda + \sqrt{w_t}} \left(\mathbb{E}_{t-1}\left[\frac{\|\boldsymbol{\xi}_t\|_2 \|\boldsymbol{g}_t\|_2}{\lambda + \sqrt{v_t}}\right] + \mathbb{E}_{t-1}\left[\frac{c \|\boldsymbol{g}_t\|_2}{\lambda + \sqrt{v_t}}\right]\right)$$
$$\overset{(c)}{\leq} \frac{\|\nabla f(\boldsymbol{x}_t)\|_2^2}{2(\lambda + \sqrt{w_t})} + \frac{1}{\lambda + \sqrt{w_t}} \left(\left(\mathbb{E}_{t-1}\left[\frac{\|\boldsymbol{\xi}_t\|_2 \|\boldsymbol{g}_t\|_2}{\lambda + \sqrt{v_t}}\right]\right)^2 + \left(\mathbb{E}_{t-1}\left[\frac{c \|\boldsymbol{g}_t\|_2}{\lambda + \sqrt{v_t}}\right]\right)^2\right), \tag{31}$$

where $(a)$ is by $\frac{\|\nabla f(\boldsymbol{x}_t)\|_2}{\lambda + \sqrt{w_t}} \in \mathcal{F}_{t-1}$, $(b)$ is due to $\|\boldsymbol{g}_t\|_2 + \|\nabla f(\boldsymbol{x}_t)\|_2 \leq \sqrt{w_t} + \sqrt{v_t}$ and $c \leq \sqrt{w_t} + \sqrt{v_t}$, and $(c)$ holds by AM-GM inequality, i.e., $\|\nabla f(\boldsymbol{x}_t)\|_2 X \leq \frac{\|\nabla f(\boldsymbol{x}_t)\|_2^2}{4} + X^2$ for $X = \mathbb{E}_{t-1}\left[\frac{\|\boldsymbol{\xi}_t\|_2 \|\boldsymbol{g}_t\|_2}{\lambda + \sqrt{v_t}}\right]$ and $\mathbb{E}_{t-1}\left[\frac{c\|\boldsymbol{g}_t\|_2}{\lambda + \sqrt{v_t}}\right]$, respectively. Next, we apply Hölder's inequality to get

$$\left(\mathbb{E}_{t-1}\left[\frac{\|\boldsymbol{\xi}_t\|_2 \|\boldsymbol{g}_t\|_2}{\lambda + \sqrt{v_t}}\right]\right)^2 \leq \left(\mathbb{E}_{t-1}\left[\|\boldsymbol{\xi}_t\|_2^p\right]\right)^{\frac{2}{p}} \left(\mathbb{E}_{t-1}\left[\frac{\|\boldsymbol{g}_t\|_2^{\bar{p}}}{(\lambda + \sqrt{v_t})^{\bar{p}}}\right]\right)^{\frac{2}{\bar{p}}}$$
$$\overset{\|\cdot\|_2 \leq \|\cdot\|_p, \text{Assumption } 2.4}{\leq} \|\boldsymbol{\sigma}\|_p^2 \left(\mathbb{E}_{t-1}\left[\frac{\|\boldsymbol{g}_t\|_2^{\bar{p}}}{(\lambda + \sqrt{v_t})^{\bar{p}}}\right]\right)^{\frac{2}{\bar{p}}} \leq \|\boldsymbol{\sigma}\|_p^2 \left(\mathbb{E}_{t-1}\left[\left(\frac{\|\boldsymbol{g}_t\|_2^2}{\lambda^2 + v_t}\right)^{\frac{\bar{p}}{2}}\right]\right)^{\frac{2}{\bar{p}}}, \tag{32}$$

and

$$\left(\mathbb{E}_{t-1}\left[\frac{c\|\boldsymbol{g}_t\|_2}{\lambda + \sqrt{v_t}}\right]\right)^2 \leq \mathbb{E}_{t-1}\left[\frac{c^2 \|\boldsymbol{g}_t\|_2^2}{(\lambda + \sqrt{v_t})^2}\right] \leq \mathbb{E}_{t-1}\left[\frac{c^2 \|\boldsymbol{g}_t\|_2^2}{\lambda^2 + v_t}\right]. \tag{33}$$

Plug (32) and (33) into (31) to obtain

$$\mathbb{E}_{t-1}\left[\left\langle \nabla f(\boldsymbol{x}_t), \frac{\boldsymbol{g}_t}{\lambda + \sqrt{w_t}} - \frac{\boldsymbol{g}_t}{\lambda + \sqrt{v_t}} \right\rangle\right]$$
$$\leq \frac{\|\nabla f(\boldsymbol{x}_t)\|_2^2}{2(\lambda + \sqrt{w_t})} + \frac{\|\boldsymbol{\sigma}\|_p^2}{\lambda + \sqrt{w_t}} \left(\mathbb{E}_{t-1}\left[\left(\frac{\|\boldsymbol{g}_t\|_2^2}{\lambda^2 + v_t}\right)^{\frac{\bar{p}}{2}}\right]\right)^{\frac{2}{\bar{p}}} + \frac{c^2}{\lambda + \sqrt{w_t}} \mathbb{E}_{t-1}\left[\frac{\|\boldsymbol{g}_t\|_2^2}{\lambda^2 + v_t}\right]$$
$$\leq \frac{\|\nabla f(\boldsymbol{x}_t)\|_2^2}{2(\lambda + \sqrt{w_t})} + \frac{\|\boldsymbol{\sigma}\|_p^2}{c} \left(\mathbb{E}_{t-1}\left[\frac{\|\boldsymbol{g}_t\|_2^2}{\lambda^2 + v_t}\right]\right)^{\frac{2}{\bar{p}}} + c\mathbb{E}_{t-1}\left[\frac{\|\boldsymbol{g}_t\|_2^2}{\lambda^2 + v_t}\right], \tag{34}$$

where the last step is by $\lambda + \sqrt{w_t} \geq c$, $\frac{\|\boldsymbol{g}_t\|_2^2}{\lambda^2 + v_t} \leq 1$, and $\frac{\bar{p}}{2} \geq 1$.

We combine (30) and (34) to have

$$\mathbb{E}_{t-1}\left[f(\boldsymbol{x}_{t+1})\right] \leq f(\boldsymbol{x}_t) - \frac{\gamma\|\nabla f(\boldsymbol{x}_t)\|_2^2}{2(\lambda + \sqrt{w_t})} + \gamma\frac{\|\boldsymbol{\sigma}\|_p^2}{c}\left(\mathbb{E}_{t-1}\left[\frac{\|\boldsymbol{g}_t\|_2^2}{\lambda^2 + v_t}\right]\right)^{\frac{2}{p}} + \gamma\left(c + \frac{\gamma\|\boldsymbol{L}\|_\infty}{2}\right)\mathbb{E}_{t-1}\left[\frac{\|\boldsymbol{g}_t\|_2^2}{\lambda^2 + v_t}\right].$$

Taking expectations on both sides and rearranging terms, we know

$$\frac{\gamma}{2}\mathbb{E}\left[\frac{\|\nabla f(\boldsymbol{x}_t)\|_2^2}{\lambda + \sqrt{w_t}}\right] \leq \mathbb{E}\left[f(\boldsymbol{x}_t) - f(\boldsymbol{x}_{t+1})\right] + \gamma\frac{\|\boldsymbol{\sigma}\|_p^2}{c}\mathbb{E}\left[\left(\mathbb{E}_{t-1}\left[\frac{\|\boldsymbol{g}_t\|_2^2}{\lambda^2 + v_t}\right]\right)^{\frac{2}{p}}\right] + \gamma\left(c + \frac{\gamma\|\boldsymbol{L}\|_\infty}{2}\right)\mathbb{E}\left[\frac{\|\boldsymbol{g}_t\|_2^2}{\lambda^2 + v_t}\right].$$

Finally, noticing that $\frac{\bar{p}}{2} \geq 1$, we hence can invoke Hölder's inequality again to have

$$\mathbb{E}\left[\left(\mathbb{E}_{t-1}\left[\frac{\|\boldsymbol{g}_t\|_2^2}{\lambda^2 + v_t}\right]\right)^{\frac{2}{\bar{p}}}\right] \leq \left(\mathbb{E}\left[\mathbb{E}_{t-1}\left[\frac{\|\boldsymbol{g}_t\|_2^2}{\lambda^2 + v_t}\right]\right]\right)^{\frac{2}{\bar{p}}} = \left(\mathbb{E}\left[\frac{\|\boldsymbol{g}_t\|_2^2}{\lambda^2 + v_t}\right]\right)^{\frac{2}{\bar{p}}},$$

which leads us to the desired result. $\qquad\square$

**Lemma C.3.** *For any $T \in \mathbb{N}$ and $q \in [0, 1]$,* `AdaGrad-Norm` *(Algorithm 2) guarantees*

$$\sum_{t=1}^{T}\left(\mathbb{E}\left[\frac{\|\boldsymbol{g}_t\|_2^2}{\lambda^2 + v_t}\right]\right)^q \leq T^{1-q}\left(\mathbb{E}\left[\ln\left(1 + \frac{v_T}{\lambda^2}\right)\right]\right)^q.$$

*Proof.* By the concavity of $x^q$ (since $q \in [0, 1]$), we have

$$\frac{1}{T}\sum_{t=1}^{T}\left(\mathbb{E}\left[\frac{\|\boldsymbol{g}_t\|_2^2}{\lambda^2 + v_t}\right]\right)^q \leq \left(\frac{1}{T}\sum_{t=1}^{T}\mathbb{E}\left[\frac{\|\boldsymbol{g}_t\|_2^2}{\lambda^2 + v_t}\right]\right)^q,$$

which implies that

$$\sum_{t=1}^{T}\left(\mathbb{E}\left[\frac{\|\boldsymbol{g}_t\|_2^2}{\lambda^2 + v_t}\right]\right)^q \leq T^{1-q}\left(\mathbb{E}\left[\sum_{t=1}^{T}\frac{\|\boldsymbol{g}_t\|_2^2}{\lambda^2 + v_t}\right]\right)^q = T^{1-q}\left(\mathbb{E}\left[\sum_{t=1}^{T}1 - \frac{\lambda^2 + v_{t-1}}{\lambda^2 + v_t}\right]\right)^q$$

$$\overset{(a)}{\leq} T^{1-q}\left(\mathbb{E}\left[\sum_{t=1}^{T}\ln\left(\frac{\lambda^2 + v_t}{\lambda^2 + v_{t-1}}\right)\right]\right)^q = T^{1-q}\left(\mathbb{E}\left[\ln\left(1 + \frac{v_T}{\lambda^2}\right)\right]\right)^q,$$

where $(a)$ is due to $1 - x^{-1} \leq \ln x, \forall x > 0$. $\qquad\square$

**Lemma C.4** (Restatement of Lemma 4.7). *Under Assumption 2.4, for any $t \in \mathbb{N}$,* `AdaGrad-Norm` *(Algorithm 2) guarantees*

$$\mathbb{E}\left[\sqrt{v_t}\right] \leq \sqrt{2}\|\boldsymbol{\sigma}\|_p t^{\frac{1}{p}} + \mathbb{E}\left[\sqrt{2u_t}\right],$$

*where $u_t \triangleq \sum_{s=1}^{t}\|\nabla f(\boldsymbol{x}_s)\|_2^2, \forall t \in \mathbb{N}$.*

**Lemma C.5.** *Under Assumptions 2.2 and 2.4,* `AdaGrad-Norm` *(Algorithm 2) guarantees*

$$\mathbb{E}\left[\ln\left(1 + \frac{v_T}{\lambda^2}\right)\right] \leq 2\ln\left(1 + \frac{\sqrt{2}\|\boldsymbol{\sigma}\|_p T^{\frac{1}{p}} + \mathbb{E}\left[\sqrt{2u_T}\right]}{\lambda}\right) \leq 2\ln K_T,$$

*where $K_T \triangleq 1 + \frac{\sqrt{2}\|\boldsymbol{\sigma}\|_p T^{\frac{1}{p}} + 2\|\nabla f(\boldsymbol{x}_1)\|_2 T^{\frac{1}{2}} + 2\gamma\|\boldsymbol{L}\|_\infty T^{\frac{3}{2}}}{\lambda}$.*

*Proof.* Note that

$$\mathbb{E}\left[\ln\left(1+\frac{v_T}{\lambda^2}\right)\right] = 2\mathbb{E}\left[\ln\left(\sqrt{1+\frac{v_T}{\lambda^2}}\right)\right] \leq 2\mathbb{E}\left[\ln\left(1+\frac{\sqrt{v_T}}{\lambda}\right)\right] \leq 2\ln\left(1+\frac{\mathbb{E}\left[\sqrt{v_T}\right]}{\lambda}\right),$$

where the last step is due to the concavity of $\ln x$. Next, we invoke Lemma C.4 to obtain

$$\mathbb{E}\left[\ln\left(1+\frac{v_T}{\lambda^2}\right)\right] \leq 2\ln\left(1+\frac{\sqrt{2}\left\|\boldsymbol{\sigma}\right\|_p T^{\frac{1}{p}} + \mathbb{E}\left[\sqrt{2u_T}\right]}{\lambda}\right). \tag{35}$$

Moreover, under Assumption 2.2, we have almost surely, for any $t \in [T]$,

$$\left\|\nabla f(\boldsymbol{x}_t) - \nabla f(\boldsymbol{x}_1)\right\|_{1/\boldsymbol{L}} \leq \left\|\boldsymbol{x}_t - \boldsymbol{x}_1\right\|_{\boldsymbol{L}} \leq \sum_{s=1}^{t-1}\left\|\boldsymbol{x}_{s+1} - \boldsymbol{x}_s\right\|_{\boldsymbol{L}}$$

$$= \sum_{s=1}^{t-1}\frac{\gamma\left\|\boldsymbol{g}_s\right\|_{\boldsymbol{L}}}{\lambda + \sqrt{v_s}} \leq \sum_{s=1}^{t-1}\gamma\sqrt{\left\|\boldsymbol{L}\right\|_\infty} = \gamma\sqrt{\left\|\boldsymbol{L}\right\|_\infty}(t-1)$$

$$\Rightarrow \left\|\nabla f(\boldsymbol{x}_t) - \nabla f(\boldsymbol{x}_1)\right\| \leq \gamma\left\|\boldsymbol{L}\right\|_\infty(t-1).$$

Hence, there is almost surely

$$\sqrt{2u_T} = \sqrt{2\sum_{t=1}^{T}\left\|\nabla f(\boldsymbol{x}_t)\right\|_2^2} \leq \sqrt{2\sum_{t=1}^{T}\left(\left\|\nabla f(\boldsymbol{x}_1)\right\|_2 + \gamma\left\|\boldsymbol{L}\right\|_\infty(t-1)\right)^2}$$

$$\leq 2\sqrt{\sum_{t=1}^{T}\left\|\nabla f(\boldsymbol{x}_1)\right\|_2^2 + \sum_{t=1}^{T}\gamma^2\left\|\boldsymbol{L}\right\|_\infty^2(t-1)^2}$$

$$\leq 2\left\|\nabla f(\boldsymbol{x}_1)\right\|_2 T^{\frac{1}{2}} + 2\gamma\left\|\boldsymbol{L}\right\|_\infty T^{\frac{3}{2}}. \tag{36}$$

Finally, we plug (36) back into (35) to have

$$2\ln\left(1+\frac{\sqrt{2}\left\|\boldsymbol{\sigma}\right\|_p T^{\frac{1}{p}} + \mathbb{E}\left[\sqrt{2u_T}\right]}{\lambda}\right) \leq 2\ln\left(1+\frac{\sqrt{2}\left\|\boldsymbol{\sigma}\right\|_p T^{\frac{1}{p}} + 2\left\|\nabla f(\boldsymbol{x}_1)\right\|_2 T^{\frac{1}{2}} + 2\gamma\left\|\boldsymbol{L}\right\|_\infty T^{\frac{3}{2}}}{\lambda}\right) = 2\ln K_T.$$

$\square$

