# OpenReview forum: "Can Adaptive Gradient Methods Converge under Heavy-Tailed Noise? A Case Study of AdaGrad"
_ICML.cc/2026/Conference — ICML 2026 regular_

### Official Review · Reviewer_zkoj · 2026-03-01

**Soundness:** 3
**Presentation:** 3
**Significance:** 2
**Originality:** 3
**Overall Recommendation:** 4
**Confidence:** 5

**Summary:**

This paper provided a rigorous theoretical analysis of AdaGrad and its variant, AdaGrad-Norm under heavy-tailed noise.  Their results are adaptive to noise level, and the lower bound of AdaGrad established in this paper is also tighter than existing literature. The new decorrelated stepsizes potentially play an important role in deriving these theoretical results.

**Compliance With Llm Reviewing Policy:**

Affirmed.

**Final Justification:**

The theoretical contribution of this paper is not bad, although there are some overclaims on evaluating existing literature and restrictive assumptions on the coordinate-wise noise for AdaGrad-Norm. The authors have revised the overclaimed points and provide a convincing explanation of the assumption.

I think the current form of this paper can be published in ICML. I recommend to accept this paper.

**Key Questions For Authors:**

Please reply to the "**Weakness**" stated above.

**Limitations:**

yes

**Strengths And Weaknesses:**

Strengths:

(1) The theoretical contribution is clear. The summary of theoretical contribution refers to "**Summary**".

(2) The new proposed decorrelated stepsizes potentially provide new insights for analyzing other algorithms with adaptive stepsizes.

Weakness:

(1) In my opinion, the main weakness in this paper is evaluating delayed AdaGrad not very accurate. The authors claim that delayed adaptive stepsizes simplify theoritical analysis, since it makes the stepsize and the stochastic gradient conditionally independent. In fact, the delayed and non-delayed adaptive stepsizes has corresponding difficulties, respectively. As pointed out in [Li and Orabona, 2019], one can not simply bound $\mathbb{E}[\gamma_t^2||g_t||^2]$ for the delayed adaptive stepsizes, while it can be bounded directly by non-delayed adaptive stepsizes. This overclaim harms  the importance of theoretical results in this paper.

(2) The authors made heavy-tailed noise assumption in a coordinate-wise form. I agree this assumption for the coordinate-wise update AdaGrad. However, this coordinate-wise form of heavy-tailed noise assumption is not necessary for AdaGrad-Norm. This restrictive assumption on AdaGrad-Norm potentially decreases the persuasion of the AdaGrad-Norm's results.

**Additional point**: It is better for the authors to provide numerical experiments on real-world datasets, since the heavy-tailed noise setting also possess great practical meanings, not only in theoretical aspects. Note that this point is **not** the weakness of this paper, which is only the reviewer's suggestion.

**Summary of Evaluation:** This paper has some theoretical contributions, while has some potential overclaim and several points can be improved (mainly lies in (1) and (2) in "**Weakness**"). I think the current version of this paper is approximately the borderline accept level. I belive that major revision of this paper  can significantly improve the quality of this paper. I am OK if the area chair finally decided to
accept this paper. Also, it is possible that I miss something and I am open to change my mind.

---

> ### Author Rebuttal · Authors · 2026-03-29
>
> We appreciate the reviewer's valuable feedback. Below, we answer the reviewer's questions.
>
> **W1.** Thanks for the suggestion. We have rephrased the corresponding statement in accordance with the reviewer's advice. The modification will be reflected in the revision.
>
> **W2.** Thanks for the comment. We would like to make two clarifications here:
>
> 1. The coordinate-wise heavy-tailed assumption (Assumption 2.4 in the paper) is indeed unnecessary for AdaGrad-Norm. Instead, one can assume the common norm-based version $\mathbb{E}\_{t-1}[\Vert\boldsymbol{\xi}\_t\Vert_2^p]\leq\sigma^p$ for some $\sigma\geq0$ (let us call this condition **Assumption 2.4'** for convenience). This fact can be seen from the current analysis for AdaGrad-Norm in Section 4 and Appendix C. Concretely, the proof for AdaGrad-Norm in these two sections only depends on the quantity $\Vert\boldsymbol{\sigma}\Vert_p$, which can be recognized as the parameter $\sigma$ in Assumption 2.4' (see below for why).  Hence, Assumption 2.4 actually does not weaken the results of AdaGrad-Norm, as the current analysis extends beyond this assumption and is applicable to Assumption 2.4', provably. The reason that we retain the coordinate-wise assumption in the analysis of AdaGrad-Norm is for simplicity, aiming to avoid introducing new assumptions.
>
> 1. In addition, we would like to mention that the current Assumption 2.4 (i.e., $\mathbb{E}\_{t-1}[|\boldsymbol{\xi}\_{t,i}|^p]\leq\boldsymbol{\sigma}\_i^p,\forall i\in[d]$) and the new Assumption 2.4' mentioned above (i.e., $\mathbb{E}\_{t-1}[\Vert\boldsymbol{\xi}\_t\Vert_2^p]\leq\sigma^p$) are in fact “interchangeable”. Precisely, they are equivalent in the following sense.
>
>    - Assumption 2.4 holds $\Rightarrow$ Assumption 2.4' holds with $\sigma=\Vert\boldsymbol{\sigma}\Vert_p$. This is true due to $\mathbb{E}\_{t-1}[\Vert\boldsymbol{\xi}\_t\Vert_2^p]\overset{(a)}{\leq}\mathbb{E}\_{t-1}[\Vert\boldsymbol{\xi}\_t\Vert_{p}^p]\overset{(b)}{\leq}\sum_{i=1}^d\boldsymbol{\sigma}\_i^p=\Vert\boldsymbol{\sigma}\Vert_p^p$, where $(a)$ is by $\Vert\cdot\Vert_2\leq\Vert\cdot\Vert_p,\forall p\in[1,2]$ and $(b)$ is by Assumption 2.4.
>
>    - Assumption 2.4' holds $\Rightarrow$ Assumption 2.4 holds with some $\boldsymbol{\sigma}\in\mathbb{R}\_{\geq0}^d$. This is true since for each $i\in[d]$, $\mathbb{E}\_{t-1}[|\boldsymbol{\xi}\_{t,i}|^p]\leq\mathbb{E}\_{t-1}[\Vert\boldsymbol{\xi}\_t\Vert_2^p]<+\infty$, which implies that there must exist $\boldsymbol{\sigma}\_i\in\mathbb{R}\_{\geq0}$ such that $\mathbb{E}\_{t-1}[|\boldsymbol{\xi}\_{t,i}|^p]\leq\boldsymbol{\sigma}\_i^p$, meaning that Assumption 2.4 must hold for some $\boldsymbol{\sigma}\in\mathbb{R}\_{\geq0}^d$.

---

> > ### Author Rebuttal · Reviewer_zkoj · 2026-04-01
> >
> > Thank you for your response. My concerns are almost addressed. I will maintain my positive evaluation.

---

### Official Review · Reviewer_ibnj · 2026-03-10

**Soundness:** 3
**Presentation:** 2
**Significance:** 4
**Originality:** 3
**Overall Recommendation:** 4
**Confidence:** 3

**Summary:**

This paper studies the convergence behavior of adaptive gradient methods under heavy-tailed gradient noise. While recent theoretical works have established convergence guarantees for stochastic optimization under heavy-tailed noise using techniques such as gradient clipping or normalization, these results do not cover widely used adaptive optimizers such as AdaGrad or Adam.

The paper focuses on AdaGrad as a case study and analyzes its convergence in non-convex optimization under heavy-tailed noise, where the noise only has a finite $p$-th moment for $p\in(1,2]$. The authors establish the first convergence rate for AdaGrad in this setting when $p>4/3$, without requiring prior knowledge of the tail index. The analysis shows that AdaGrad can adapt to the tail index and noise level automatically.

In addition, the paper derives an algorithm-dependent lower bound for AdaGrad under heavy-tailed noise, showing that AdaGrad cannot achieve the minimax optimal rate in general. The paper also analyzes a variant, AdaGrad-Norm, and shows that it can achieve an improved convergence rate under an additional bounded-objective assumption.

Overall, the paper aims to provide theoretical understanding of why adaptive gradient methods may remain stable under heavy-tailed gradient noise.

**Compliance With Llm Reviewing Policy:**

Affirmed.

**Key Questions For Authors:**

See weaknesses 2.

**Limitations:**

Yes

**Strengths And Weaknesses:**

# Strengths

**1. Important and timely research question**

The paper studies the robustness of adaptive gradient methods under heavy-tailed noise, which is a relevant problem for modern deep learning optimization. Empirical studies have observed heavy-tailed gradient noise during neural network training, and understanding why adaptive optimizers remain stable in such settings is an interesting theoretical question.

**2. First convergence result for AdaGrad under heavy-tailed noise**

The paper provides the first convergence guarantee for AdaGrad in non-convex optimization with heavy-tailed noise when $p>4/3$. The result does not require prior knowledge of the tail index and shows that AdaGrad can adapt automatically to the noise level and tail behavior.

**3. Algorithm-dependent lower bound**

The paper derives an algorithm-dependent lower bound that explicitly captures the dependence on the learning rate. This contributes additional theoretical insight into the limitations of AdaGrad in heavy-tailed settings.

**4. Technically solid analysis**

The analysis introduces a modified proxy stepsize technique and carefully handles heavy-tailed noise in the convergence proof. The theoretical development appears technically nontrivial and builds on recent literature in heavy-tailed stochastic optimization.

------

# Weaknesses

**1. Limited scope (AdaGrad only)**

The analysis focuses solely on AdaGrad, while modern deep learning practice predominantly uses Adam or AdamW. As a result, the practical impact of the results may be somewhat limited. Extending the analysis to more widely used adaptive optimizers would significantly strengthen the contribution.

**2. Lack of empirical evidence**

The paper does not include empirical experiments or empirical analysis of gradient noise distributions in practical training settings. Even a small empirical study (e.g., estimating tail indices during neural network training) would help clarify the practical relevance of the theoretical assumptions.

---

> ### Author Rebuttal · Authors · 2026-03-29
>
> We appreciate the reviewer's valuable feedback. Below, we answer the reviewer's questions.
>
> **W1.** We believe that our idea of the generalized proxy stepsize (i.e., imposing a free parameter $\boldsymbol{c}\in\mathbb{R}_{\geq 0}^d$) can be extended at least to Adam. This is because, as far as we know, the existing convergence proof for Adam also relies on the proxy stepsize technique, but without this free parameter (for example, see [1]). Therefore, by combining our new idea with a careful analysis, it is highly possible to obtain new convergence results for Adam under heavy-tailed noise, which we view as a promising direction for future work.
>
> **W2.** We did not include additional empirical experiments to estimate the tail index because many prior works in the literature have already investigated it and observed the heavy-tailed phenomenon, for example, see [2-6]. We are happy to mention these existing empirical evidence in the revision to clarify the practical relevance of the theoretical assumptions.
>
> **References**
>
> [1] Wang, Bohan, et al. Closing the Gap Between the Upper Bound and the Lower Bound of Adam’s Iteration Complexity. NeurIPS 2023.
>
> [2] Simsekli, Umut, et al. A Tail-Index Analysis of Stochastic Gradient Noise in Deep Neural Networks. ICML 2019.
>
> [3] Zhang, Jingzhao, et al. Why are Adaptive Methods Good for Attention Models? NeurIPS 2020.
>
> [4] Garg, Saurabh, et al. On Proximal Policy Optimization’s Heavy-tailed Gradients. ICML 2021.
>
> [5] Battash, Barak, et al. Revisiting the Noise Model of Stochastic Gradient Descent. AISTATS 2024.
>
> [6] Ahn, Kwangjun, et al. Linear attention is (maybe) all you need (to understand transformer optimization). ICLR 2024.

---

> > ### Author Rebuttal · Reviewer_ibnj · 2026-04-01
> >
> > Thank you for your response. I will maintian my positive score to support your publication.

---

### Official Review · Reviewer_xry8 · 2026-03-12

**Soundness:** 4
**Presentation:** 4
**Significance:** 4
**Originality:** 4
**Overall Recommendation:** 5
**Confidence:** 4

**Summary:**

The paper focuses primarily on AdaGrad and provides what appears to be the first convergence guarantee for standard AdaGrad in non-convex heavy-tailed optimization. The main result shows a convergence rate of $O(T^{-(3p-4)/(4p)})$ for tail index $p\in(4/3,2]$, without requiring prior knowledge of $p$. This adaptivity is a central conceptual contribution, since avoiding problem-dependent tuning is one of the main appeals of adaptive methods. The paper also develops an algorithm-dependent lower bound for AdaGrad, explicitly reflecting the dependence on the input learning rate, and uses it to argue that the known minimax heavy-tailed rates are generally not attainable by AdaGrad. This lower bound is positioned as both new in the heavy-tailed setting and stronger than prior lower bounds even in the $p=2$ case. Finally, the paper studies AdaGrad-Norm. Under an additional bounded-objective assumption, it proves an improved rate $O(T^{-(p-1)/(2p)})$ for all $p\in(1,2]$. Without the bounded-objective assumption, the paper provides a rate matching the AdaGrad result in its dependence on $T$.

**Compliance With Llm Reviewing Policy:**

Affirmed.

**Final Justification:**

This is a strong paper with interesting topic, rigorous derivation, and insightful results. Therefore, my final score is 5 - Accept, to support its publication.

**Key Questions For Authors:**

(1) How should the readers interpret the main AdaGrad rate $O(T^{-(3p-4)/(4p)})$?, compared to the known minimax rates for general heavy-tailed non-convex optimization? A concise comparison table or discussion would be helpful;

(2) Is the gap between the AdaGrad upper and lower bounds mainly due to the limitation of the proof techniques, or is there anything even more interesting that results in such a limitation in AdaGrad? What optimal rate would you conjecture for AdaGrad in this setting?

(3) Do the proofs rely critically on coordinate-wise adaption, or could some ideas extend to Adam/AdamW-style momentum?

**Limitations:**

Yes.

**Strengths And Weaknesses:**

Strengths:

(1) The paper addresses a real and timely gap between theory and practice. Heavy-tailed gradient noise is frequently encountered in ML, while the current literature in theory hasn't fully solved the convergence result;

(2) The motivation is very clear. The authors carefully explained why the existing results, especially those for clipped/normalized SGD and the recent work of Chezhegov et al. (2025), do not yest provide a good answer for standard adaptive methods;

(3) The theoretical contribution is strong, and the mathematical proofs appear to be solid. All the assumptions and obtained rates are well justified;

(4) My favorite part is the lower bound that is algorithm dependent, which helps clarify what AdaGrad can and cannot achieve.

Weaknesses:

I only have good things to say about this paper. If I have to mention one, then I would suggest adding some simulation studies to verify the rates derived theoretically. But I do understand that this paper focuses on theoretical contributions, so it is fine with me to leave it as it is.

---

> ### Author Rebuttal · Authors · 2026-03-29
>
> We appreciate the reviewer's valuable feedback. Below, we answer the reviewer's questions.
>
> **Q1.** To prove the minimax rate $\Theta(1/T^{\frac{p-1}{3p-2}})$, people always assume the tail index $p$ is known and fixed (for both upper and lower bounds). In contrast, our $O(1/T^{\frac{3p-4}{4p}})$ rate for AdaGrad should be interpreted as a result that is oblivious, and hence adaptive to, the unknown index $p$. The difference between these two rates could be viewed as a trade-off in terms of prior information about the tail index $p$. We are happy to include a table in the revision, as suggested by the reviewer, if more space is available.
>
> **Q2.** This is a good question that we have also spent some time thinking about. We summarize our current thoughts as follows.
>
> - Upper bound: For AdaGrad, we have no idea whether Theorem 3.1 is tight or not. For AdaGrad-Norm under bounded functions, we incline to say that our Theorem 4.1 is tight, as it matches the best possible parameter-free rate for normalized gradient methods. (Though, of course, these two kinds of methods are not directly comparable in general.) Finally, if our current upper bound for AdaGrad is loose, we conjecture that the tight rate is of the same order as $O(1/T^{\frac{p-1}{2p}})$ in Theorem 4.1.
>
> - Lower bound: We suspect that Theorem 3.3 is not tight in $\epsilon$ (as mentioned in Lines 317-320), possibly due to limitations in the proof techniques. This is because our proof builds upon the framework developed by two prior works [1] and [2], which was however designed for problems different from the one studied in our paper. This means that, even after further modifications and refinements in our work, the high-level idea underlying the analysis may not perfectly fit our problem. Therefore, we believe there is still room to improve the lower bound.
>
> **Q3.** We believe that our idea of the generalized proxy stepsize (i.e., imposing a free parameter $\boldsymbol{c}\in\mathbb{R}_{\geq0}^d$) can be extended at least to Adam. This is because, as far as we know, the existing convergence proof for Adam also relies on the proxy stepsize technique, but without this free parameter (for example, see [3]). Therefore, by combining our new idea with a careful analysis, it is highly possible to obtain new convergence results for Adam under heavy-tailed noise, which we view as a promising direction for future work.
>
> **References**
>
> [1] Jiang, Ruichen, et al. Provable Complexity Improvement of AdaGrad over SGD: Upper and Lower Bounds in Stochastic Non-Convex Optimization. COLT 2025.
>
> [2] Hübler, Florian, et al. From Gradient Clipping to Normalization for Heavy Tailed SGD. AISTATS 2025.
>
> [3] Wang, Bohan, et al. Closing the Gap Between the Upper Bound and the Lower Bound of Adam’s Iteration Complexity. NeurIPS 2023.

---

> > ### Author Rebuttal · Reviewer_xry8 · 2026-03-31
> >
> > Thank you very much for carefully addressing my questions. I greatly appreciate your conjecture on the tightness of your main results. I will keep my positive score to support your publication.

---

### Decision · Program_Chairs · 2026-04-30

**Decision:**

Accept (regular)

**Comment:**

The reviewers found that the paper addresses an important and timely problem in stochastic optimisation, namely the convergence of stochastic adaptive optimization methods under heavy-tailed gradient noise. They agree that the work contains significantly novel technical contributions.

While some concerns were raised regarding the lack of proper numerical investigation, the strength and novelty of the theoretical contributions largely mitigate this shortcoming. Additional questions and concerns were identified during the review process, which the authors have addressed in the rebuttal to the satisfaction of the reviewers.

The authors are strongly encouraged to incorporate the suggested amendments and improvements discussed during the rebuttal into the final version of the paper.